

# Temporal variability of chlorophyll distribution in the Gulf of Mexico: bio-optical data from profiling floats

Orens Pasqueron de Fommervault[1], Paula Perez-Brunius[1], Pierre Damien[1], Julio Sheinbaum[1]

[1] Departamento de Oceanografìa Fisica, Centro de Investigacion Cientìfica y de Educacion Superior de Ensenada, Ensenada, 22860, Mexico

*Correspondence to*: Orens Pasqueron de Fommervault (orens@cicese.mx)

**Abstract.** Chlorophyll concentration is a key oceanic biogeochemical variable. In the Gulf of Mexico (GOM), its distribution, which is mainly obtained from satellite surface observations and scarce *in situ* experiments, is still poorly understood. In 2011-2012, eight profiling floats equipped with biogeochemical sensors were deployed for the first time in the GOM and generated an unprecedented dataset that significantly increased the number of chlorophyll vertical distribution measurements in the region. The analysis of these data, once calibrated, permits us to reconsider the spatial and temporal variability of the chlorophyll concentration in the water column. At a seasonal scale, results confirm the surface signal seen by satellites, presenting maximum concentrations in winter and low values in summer. It is shown that the deepening of the mixed layer depth is the primary factor triggering the chlorophyll surface increase in winter. In the GOM, current belief is that this surface increase corresponds to a biomass increase. However, the present dataset reveals a vertically integrated content of chlorophyll which remains constant throughout the year, suggesting that the surface increase results from a vertical redistribution of subsurface chlorophyll or photoacclimation processes, rather than a net increase of primary productivity. One plausible explanation for this is the decoupling between the mixed layer depth and the deep nutrient reservoir since mixed layer depth only reaches the nitracline in sporadic events in the observations. Float measurements also provide evidence that the depth and the magnitude of the deep chlorophyll maximum is strongly controlled by the mesoscale variability, with higher chlorophyll biomass generally observed in cyclones rather than anticyclones.

## 1 Introduction

The Gulf of Mexico (GOM) is a semi-enclosed sea linked to the Caribbean Sea through the Yucatan Channel and to the Atlantic Ocean through the Florida Straits. It is characterized by a complex circulation dominated by the Loop Current (LC) that regularly forms large anticyclonic eddies (∼ 200-300km diameter) that propagate westwards (e.g. Sturges and Leben, 2000). LC and LC eddies can be identified by Caribbean subtropical underwater (*i.e.* high salinity in the upper layer), and are clearly distinguishable from Gulf of Mexico common water which is formed by vertical convective mixing within the gulf's uppermost 200 m in winter or by mixing induced by the collision of LC rings against the western gulf boundary (Elliott, 1982; Nowlin and McClellan, 1967; Vidal et al., 1992). Another important feature of the circulation in the GOM is the presence of relatively



smaller cyclonic and anticyclonic eddies in many parts of the basin (e.g. Hamilton et al., 2002; Hamilton, 2007; Schmitz, 2005).

From a biogeochemical point of view, the GOM, as a whole, is a moderately high productivity ecosystem ($< 300$ gC m$^{-2}$ yr$^{-1}$) with a contrasting trophic environment (Heileman and Rabalais, 2009). The deep basin and the continental shelf are considered

oligotrophic and nutrient-limited being relatively isolated from coastal and eutrophic waters (Heileman and Rabalais, 2009). *In situ* measurements indicate surface chlorophyll concentration (hereafter [CHL]$_{surf}$) ranging from 0.06 to 0.32 mg m$^{-3}$, with [CHL] 2-3 times higher in subsurface waters (Biggs and Ressler, 2001).

Other studies, mostly based on satellite surface chlorophyll measurements (Müller-Karger et al., 1991; Salmerón-García et al., 2011), or numerical simulations (Fennel et al., 2011; Walsh et al., 1989; Xue et al., 2013), suggest important seasonal

variations. Lower [CHL]$_{surf}$ values are observed from May to July and maximum values are found in winter from December to January. This cycle is considered to be primarily triggered by annual changes in ocean-atmosphere fluxes (Virmani and Weisberg, 2003), resulting in the deepening of the mixed layer in winter. In the GOM, it is currently thought that a higher [CHL]$_{surf}$ in winter occurs concomitantly with a biomass increase, as a consequence of nutrient entrainment to the surface (Jolliff et al., 2008; Melo Gonzalez et al., 2000; Müller-Karger et al., 1991; Müller-Karger et al., 2015; Salmerón-García et al.,

2011). However, this has never been truly demonstrated due to the absence of sufficient data at the proper spatio-temporal scales in the water column. Furthermore, recent studies conducted in various oligotrophic environments, also suggest that [CHL] variability in the surface layer may primarily reflect changes in intracellular pigment concentration, rather than biomass variability (Behrenfeld et al., 2005; Mignot et al., 2014; Siegel et al., 2013). Hence, the main processes giving rise to the seasonal variability of surface chlorophyll in the GOM have not yet been resolved.

Superimposed on the seasonal variability, several sporadic processes, such as mesoscale and sub-mesoscale activity (Belabbassi et al., 2005; Biggs and Ressler 2001; Linacre et al., 2015; Toner et al., 2003) or river run-off (Lohrenz et al., 1997; Nababan et al., 2011) may alter the [CHL] distribution in the deep GOM. These structures are hardly detectable from traditional *in situ* measurements (ship-based data rarely achieve the required spatio-temporal resolution), and their impact on the phytoplankton distribution and dynamics in the GOM still remains to be establisehd. The overall lack of data in the deepwater

GOM has, so far, produced a very limited picture of the [CHL] distribution, with low time and spatial resolution contributing to this uncertainty.

The development of autonomous Lagrangian platforms equipped with miniaturized bio-optical sensors now provides high-frequency and multiannual time series of physical and biogeochemical observations (Johnson et al., 2009). In the GOM, the pioneer work of Green et al., (2014) demonstrated the great potential of using profiling floats with bio-optical sensors and

showed the complex [CHL] variability present in the deep GOM. Following this first successful attempt, and with the aim to monitor the water column of the entire GOM, seven other floats with bio-optical sensors (chlorophyll and CDOM fluorimeter, Backscatterometer) and CTD were deployed in 2011 and 2012, as part of a project funded by the Bureau of Ocean Energy Management (BOEM, Hamilton et al., 2017). This floats obtained bi-weekly profiles for nearly 5 years inside the GOM.

In this study, we focus specifically on fluorescence chlorophyll (FLUO) measurements. For the first time in the GOM, we present high-frequency calibrated fluorescence chlorophyll time-series using profiling floats, simultaneously acquired in different parts of the basin. The objective is to study the influence of physical forcing on the variability of the [CHL] vertical distribution. Of particular interest is its annual cycle, since it is the first time enough *in situ* measurements are available to resolve this temporal scale at depth, and explore to what extent the surface dynamics, as seen by satellite, is representative of the variability within the water column. We also investigate the [CHL] variability at shorter time-scales and evaluate the role of mesoscale structures/water masses in shaping the annual cycle. This work provides a better understanding of the mechanisms controlling the distribution and the dynamics of phytoplankton in the deep GOM. In contrast to common belief, our analysis indicates the winter surface [CHL] maximum in the GOM is not produced by a biomass increase, but by other mechanisms described and analyzed in the following sections. This is the most important result of our study.

## 2 Data and methods

### 2.1 APEX float database

The float database is composed of eight Apex profiling floats (Teledyne-Webb Research, Inc.) deployed in the deep GOM. Two of them experienced technical issues (floats "05" and "08") and were discarded from the dataset. The profiling floats had a rest depth of 1500m and profiles were made every 14 days for most of the study. Data were transmitted in real-time using Iridium communication each time the float surfaced.

In addition to the standard conductivity-temperature-depth (CTD) sensors mounted on typical Argo profiling floats (Roemmich et al., 2003), BOEM floats were also equipped with an ECO FLbbCD-AP2 sensor (WET Labs, Inc.). This sensor allowed the measurement of [CHL] and colored dissolved organic matter (CDOM) fluorescence, as well as the optical backscattering (bbp) at 700nm (see Green et al., (2014) for more details). Bio-optical estimations were performed from 0 to 1500 m-depth (about 5 m resolution in the 0–200 m layer, 10 m resolution in the 200-500 m layer, 20 m resolution in the remaining range).

Data were collected over a time period of approximately five years (2011 through 2015), and a total of 537 profiles of both physical and bio-optical parameters were acquired in the whole GOM basin (Table 1). The resulting dataset has good spatial coverage (fig. 1, left panel), and, on a monthly basis, all periods were sampled in an equivalent manner (Fig. 1, right panel). Hereafter, we focus our discussion on the [CHL] time-series.

### 2.2 Fluorescence profiles calibration

The measurement of *in vivo* FLUO is widely used as a proxy for [CHL] (Lorenzen, 1966) which is, in turn, the main proxy for phytoplankton biomass (Cullen, 1982; Strickland, 1965). However, the conversion of FLUO into [CHL] must be done with great care. FLUO and [CHL] are generally considered proportional, which can be formalized as:



$$[CHL] = \alpha \,.\, (FLUO - \beta) \qquad\qquad\qquad\qquad\qquad\qquad (1)$$

where the $\alpha$ and $\beta$ coefficients (respectively instrumental gain and offset) are provided by the manufacturer. However, the values of these coefficients are rarely satisfactory and require post-processing evaluation (Boss et al., 2008; Guinet et al., 2013; Mignot et al., 2011; Xing et al., 2012). In addition, other biogeochemical processes, such as taxonomic composition and physiological acclimation mechanisms, are known to modulate the proportionality of FLUO and [CHL] (Cunningham et al.,1996; Falkowski and Kiefer, 1985; Kiefer, 1973), and must be taken into account for calibration purposes.

Among the physiological acclimation mechanisms affecting the FLUO-[CHL] relationship, the most serious issue is the non-photochemical quenching (NPQ) related to the decrease of the FLUO signal at the surface, in conditions of high light intensity (Cullen and Lewis, 1995). The first step of the calibration procedure is to provide a systematic correction of the NPQ. This was done following the recommendations given in the BGC-Argo quality control manual (Schmechtig et al., 2014). For this specific purpose, FLUO profiles were smoothed, using a 5-point moving median filter, and the highest FLUO value encountered above 0.9 times the mixed layer depth (MLD) was extrapolated up to the surface (Xing et al., 2012). The MLD was calculated from density profiles, using a 0.03 kg m$^{-3}$ density criterion (de Boyer Montegut et al., 2004).

The second step of the procedure is to correct FLUO profiles from instrumental offset. For each profile, the mean value at depth (*i.e.* deeper than 500m), was then computed (FLUO$_{deep}$). Assuming [CHL] is zero below 500m, $\beta$ (which indicates the response of the instrument in the absence of signal), was then determined as the median value of FLUO$_{deep}$ (Table 1).

The third step of the procedure is to evaluate the $\alpha$ parameter. This was performed following the method of Lavigne et al., (2012) and using ocean color satellite measurements (8-day composite images at 4 km spatial resolution from the Aqua MODIS satellite ocean color sensor, OCx Algorithm, available on http://oceancolor.gsfc.nasa.gov/). Float and satellite data were matched-up considering 8-day time intervals and ±0.25° spatial windows centered on the geographical position of the float profile. Corresponding satellite [CHL] values were extracted and averaged. The match-up was taken into account in the calibration procedure if at least 30% of satellite values were available. The number of valid match-ups was 60, 74, 67, 75, 82 and 74% for floats "00", "01", "02", "03", "04", and "06", respectively. The integrated chlorophyll content over 1.5 time the euphotic depth was then estimated from satellite [CHL] using empirical relationships (Uitz et al., 2006) and compared to the corresponding FLUO value (previously corrected for offset and NPQ). For a given float, $\alpha$ was finally calculated as the median value of the multiplicative coefficients obtained by the match-up (Table 1).

Although this method was not directly validated in the GOM, a comparison between satellite calibrated profiles and *in situ* HPLC [CHL] data was performed by Lavigne et al., (2012) at the station BATS (32°N and 64°W, in the Sargasso Sea). It was shown that calibrated profiles were unbiased with an associated median error of 29%, which is reduced to 16% when climatological average are compared. These values may be reasonably applied to the GOM, considering the vertical distribution of the [CHL] at BATS is relatively close to what is observed in the GOM (Michaels and Knap, 1996).





### 2.3 Particle backscattering profiles

Profiles of particulate backscattering coefficient at 700nm (bbp, in $m^{-1}$) were obtained following Green et al., (2014) protocol, and using the laboratory calibrations. High-frequency spikes were further removed from the bbp data applying successively a 5-point running median filter and a 7-point mean filter spikes, Briggs et al., 2011).

In our study, the bbp that is to first order correlated to the chlorophyll concentration (Huot et al., 2008; Loisel and Morel, 1998; Morel and Maritorena, 2001), was used as an alternative measure of the phytoplankton carbon biomass (e.g. Behrenfeld et al., 2005; Westberry et al., 2010). The [CHL]-bbp relationship is however known to be altered by physiological variations (Antoine et al., 2011; Behrenfeld and Boss, 2003) in particular because the CHL signal is strongly impacted by changes in intracellular pigment concentration resulting from photoacclimation (Fennel and Boss, 2003; Kitchen and Zaneveld, 1990; Mitchell and

Kiefer, 1988). Hence, in this work, we also considered the ratio bbp/[CHL] as a proxy of phytoplankton carbon biomass to CHL and used it to track changes in phytoplankton photoacclimation processes (Behrenfeld and Boss, 2003; Behrenfeld and Boss, 2006; Mignot et al., 2014). It is worth noting that in the absence of published empirical conversion factors in the GOM, bbp data were not converted to a carbon equivalent. We therefore considered qualitatively bbp, and used them primarily to assist in the interpretation of [CHL] profiles.

### 2.4 Nutrient data

In the present study we also used nutrients (more precisely nitrate + nitrite concentrations, hereafter [NN]) from bottle measurements acquired during the XIXIMI-2 (July 2011) and XIXIMI-3 (February-March 2013) cruises. More than 900 water samples were acquired from 74 profiles in the deep water region of the southern GOM (25°N to 20°N and 87°W to 95°W; Fig. 1, left panel).

Nutrient analyses were performed with a Skalar SANplus segmented-flow nutrient analyzer according to the protocols described in Gordon et al., (1993), and [NN] were determined according to a modification of the Armstrong et al., (1967) procedure.

### 2.5 Detection of mesoscale structures

In the present study, mesoscale structures were characterized according to the vertical distribution of temperature, considering

that isotherms are generally displaced downward (upward) in anticyclonic (cyclonic) structures in comparison with the background field (McGillicuddy and Robinson, 1997). The objective was to see if biological patterns could be identified in response to different physical situations. The classification of the eddy structures was carried out using the depth of the 6°C-isotherm (hereafter T6) for the following reasons:

-    T6 is in the lower thermocline, with a mean depth of ~795m, and is considered to separate deep stable water from the eddy dominated near surface water (Bunge et al., 2002).



- T6 exhibits a strong correlation with the upper layer eddies (Hamilton et al., 2017) and filters out a substantial part of the seasonal dynamical signal so it is a good proxy to identify eddies.

- T6 in the GOM is about 100 m deeper in anticyclones and 50 m shallower in cyclones with respect to its mean depth (Hamilton et al., 2017).

BOEM float profiles were thus gathered in different clusters according to the depth of T6, and, for statistical reasons, criteria were chosen symmetrical to the mean T6 depth to obtain groups of equivalent size. From here on, the cyclonic group (CG) will correspond to profiles with shallowest T6 (< 770 m) and the anticyclonic group (AG) to profiles of deepest T6 (> 820 m), with the remaining profiles (*i.e.* 770 m > T6 > 820 m) considered undefined or neutral. Even though T6 is not very sensitive

to seasonal variability, profiles are homogeneously distributed within each group. i.e., all seasons are equally sampled on average, with 14 profiles available per group and per month (Fig. 1, right panel). The spatial distribution of cyclones and anticyclones resulting from the T6 depth analysis are given in Fig. 2 (top panel).

A T-S diagram using all the profiles (Fig. 2, bottom right panel) shows that most cyclones have Gulf of Mexico common water (uniform salinity of ~ 36.5 between the 1024.5 and 1025.6 kg m$^{-3}$ isopycnal levels) while the anticyclones are more mixed.

Eastern structures have a clear Caribbean subtropical underwater signal (warm and salty water at the eddy's core, red diamonds on Fig. 2) whereas western structures tend towards Gulf of Mexico common water (red squares). In that sense, this classification also characterizes profiles in terms of water mass properties. One should note that our determination of mesoscale anticyclonic structures encompasses both the eddies and the Loop Current.

The relationship between T6 and sea surface height (SSH) obtained from altimetry, which is more conventionally used for the

identification of eddies in the GOM (e.g. Leben and Born, 1993), was examined in our dataset. T6 was plotted against SSH at the location and time of the profile (left bottom panel on fig. 2, see legend for SSH calculation details). Results reveal a good and positive correlation between T6 and SSH values ($R^2 = 0.58$) and confirms the results of Hamilton et al., (2017), obtained from a larger dataset (*i.e.* maximum SSH values in anticyclones and minimum values in cyclones). This also suggests that the depth of T6 is a good proxy to classify profiles from a mesoscale perspective in the GOM. The ability of the method to identify

mesoscale features was verified comparing results with those obtained using an SSH criterion which yield only a minor difference (supporting information S1).

## 3 Results and discussion

### 3.1 Seasonal cycle

During the five years of observation the BOEM floats provided a repeated coverage of the deep GOM. The mixed layer depth

(MLD, generally considered to be the main physical factor influencing upper layer phytoplankton dynamics and chlorophyll concentration ([CHL], e.g. Mann and Lazier, 2006), show consistent seasonal patterns (Fig. 3 and Fig. 4).

Regarding the MLD, in summer (*i.e.* from June to August) shallow values are measured (mean value = 18±8 m) as an indication




of a well-stratified water column. In autumn (September-October) a deepening of the mixed layer is observed, with a mean value of 37±19 m. Relative maximum values are reached in winter from December to February (mean value of 50±30 m), and this period is also characterized by a strong scattering of the values in which MLD deeper than 80 m are not uncommon. Maximum MLD are present in the float "01" time-series in winter 2012/2013 and float "06" time-series in winter 2013/2014

(Fig. 3 and Fig. 4). During these periods, maximum MLD can reach more than 150 m. In spring (from March to May), there is a gradual increase of surface density, leading to a progressive stratification of the water column and the mixed layer becomes thinner (Fig. 3 and Fig. 4).

Concerning [CHL], large variability is observed above 200 m in all time-series. Overall, a deep chlorophyll maximum (DCM), characteristic of an oligotrophic environment, is detected at around 70-100 m depth throughout the year, although this feature

tends to disappear in winter (Fig. 3, Table 2). At a seasonal scale, [CHL]$_{surf}$ here calculated as the mean [CHL] in the 0-30m layer, exhibits a clear pattern despite the strong spatial and inter-annual variability produced by mixed layer dynamics (Fig. 4). In summer, when the MLD is minimum, [CHL]$_{surf}$ is very low and generally under 0.1 mg m$^{-3}$ (Fig. 4). During this season the MLD is occasionally found deeper than the top of the DCM (defined as the first depth were [CHL]$_{surf}$ exceeds 0.1 mg m$^{-3}$, *i.e.* 4 times the detection limit of the sensor), but such events are very rare (11% of the profiles). In autumn, the mean [CHL]$_{surf}$

remains low (0.09 mg m$^{-3}$) although slightly higher than the concentration measured in summer. The MLD reaches the top of the DCM in around 40% of the autumn profiles. Maximum values of [CHL]$_{surf}$ are observed in winter (mean value of 0.22 mg m$^{-3}$) when the MLD is generally the deepest. During this season, the MLD shows large variability and generally reaches the DCM (~ 80 % of the winter profiles), which results in large dispersion of the measured [CHL]$_{surf}$ values (Fig. 4). In spring, [CHL]$_{surf}$ decreases (mean values of 0.09 mg m$^{-3}$), and a MLD deeper than the top of the DCM is only observed in 26% of the

profiles. The seasonal cycle of the [CHL]$_{surf}$ obtained from the float profiles is consistent with that reported using satellite measurements (Müller-Karger et al., 1991; Salmerón-García et al., 2011).

The [CHL]$_{surf}$ variability is, however, not mirrored by the integrated content of CHL over the 0-350 m layer ([CHL]$_{tot}$), or at least, its spatial and/or inter-annual variability is higher than the seasonal cycle. Thus, [CHL]$_{tot}$ remains almost constant all along the seasons, with a mean value around (30 mg m$^{-2}$, Fig. 4). In other words, the winter increase in [CHL]$_{surf}$ seems not to

be followed by a true biomass increase. This result is corroborated by particulate backscattering data (bbp), which could be viewed as an alternative and independent estimate of phytoplankton carbon biomass (Behrenfeld and Boss, 2003). Indeed, there is no increase of bbp in winter in either surface mean or vertical integrated values (supporting information S2). Mignot et al., (2014) concluded that such a [CHL]$_{surf}$ increase without a similar trend in the bbp signal would be the result of photoacclimation processes. They deduced this by considering separately the upper part of the photic layer (*a priori* nutrient

limited and light limited only in winter), and the lower part (*a priori* light limited but less nutrient starved). Another mechanism to explain the [CHL]$_{surf}$ increase would be a vertical redistribution of the phytoplankton over the water column (Mayot et al., 2017), given that the mixed layer is generally deep enough to reach the DCM in winter, thus connecting the upper and the lower part of the euphotic zone. This could be the case in the GOM given that in this tropical basin there is generally sufficient light to illuminate the mixed layer (Müller-Karger et al., 1991), hence sub-optimal phytoplankton growth conditions in winter





are not expected (Jolliff et al., 2008). *In situ* observations by Qian et al., (2003) suggested shifts in the surface phytoplankton community that could also account for changes in [CHL]$_{surf}$.

All these mechanisms are not necessarily exclusive, and which ever one brings to bear to explain the [CHL]$_{surf}$ increase in winter, relatively stable vertically integrated chlorophyll (and bbp) values indicate a constant phytoplankton biomass in the

water column throughout the year. The mechanistic hypothesis is that the mixed layer in winter is sufficiently deep to reach the DCM but nonetheless insufficient for bringing up nutrients. This contrasts with conclusions from previous studies conducted in the GOM that suggest an increase of biomass in winter based on surface information from satellite observations (Jolliff et al., 2008; Melo Gonzalez et al., 2000; Müller-Karger et al., 1991; Müller-Karger et al., 2015; Salmerón-García et al., 2011) and will be further discussed in section 3.3.

**3.2 Role of mesoscale structures in shaping the annual cycle**

Superimposed to a seasonal signal, float profiles also show chlorophyll variability that occurs at shorter time-scales. Subsurface temporal changes in [CHL] are closely related to isopycnals (black lines, Fig. 3), and the vertical displacement of the DCM is highly coherent with density. When profiles for which the MLD reaches the DCM are excluded (*i.e.* when the DCM structure is eroded), depth variations of the subsurface DCM are correlated with the vertical displacement of the nearby 1025.5kg m$^{-3}$-

isopycnal ($R^2 = 0.57$).

Since the 1025.5kg m$^{-3}$-isopycnal is also correlated with T6 ($R^2 = 0.48$), we observed that DCM is, on average, deeper in the anticyclonic group (AG) than in the cyclonic group (CG), whatever the time period (Table 2). Besides, a student-t test confirms that the difference in DCM mean depth observed between the two groups is statistically significant independent of season (level of significance $p = 0.05$). This variability overlaps with the seasonal deepening and shallowing of the DCM,

characterized by deepest values in summer (82±18 m in CG and 105±17 m in AG) and shallowest values in winter (68±19 m in CG and 75±11 m in AG).

To better assess the impact of the mesoscale variability on the seasonal cycle, data were monthly binned and profiles of the cyclonic (CG) and anticyclonic (AG) groups were analyzed separately. Fig. 5 shows that the MLD is generally deeper in AG than in CG, as expected (Dufois et al., 2014; Kouketsu et al., 2012), although this difference is most often not significant ($p <$

0.05, except in April, September and October) due to the strong dispersion in MLD measurements. The maximum difference is observed in January and February, with mean values around 70m in AG and 50 m in CG (Fig. 5). In both groups, maximum monthly values of [CHL]$_{surf}$ are also observed in January and February (Fig. 5). A higher increase in [CHL]$_{surf}$ is however observed in CG (only statistically significant in February), even though the mean MLD is shallower than in AG (mean [CHL]$_{surf}$ for those months range around 0.2-0.1 mg m$^{-3}$ for AG and 0.3-0.2 mg m$^{-3}$ for CG).

Consistent with the results of section 3.1, [CHL]$_{tot}$, shows no clear winter increase on either group (Fig. 5). By contrast, statistically significant differences ($p < 0.05$) between CG and AG are found from March to October, when the MLD is shallower. Monthly mean [CHL]$_{tot}$ is higher, on average, in CG (~ 32 mg m$^{-2}$) than in AG (~ 28 mg m$^{-2}$) which is most likely related to an intensification of the DCM in CG (level of significance $p = 0.05$, Table *2)*. An important question is how much



of the phytoplankton chlorophyll difference observed between CG and AG is reflective of changes in phytoplankton biomass. In particular, since the DCM is found significantly deeper in AG than in CG, one might expect that differences in $[CHL]_{tot}$, result from changes in environmental conditions (e.g. light) and a consequent modification of the ratio of CHL to phytoplankton carbon biomass (e.g. Geider, 1987). However, bbp vertical profiles show that the increase of $[CHL]_{tot}$ in CG is

also related to carbon biomass enhancement. Indeed, a higher bbp signal is observed in CG compared to AG (supporting information S3). In addition, in the DCM, the ratio bbp/CHL (proxy of phytoplankton carbon biomass to CHL), which tracks changes in phytoplankton physiology (Behrenfeld and Boss, 2003; Behrenfeld and Boss 2006; Mignot et al., 2014), is very similar between the two groups (supporting information S3). Thus, it can be reasonably assumed that the CHL difference between CG and AG in the lower euphotic zone results from biomass variations and not from photoacclimation processes. As

a consequence, in the GOM, the phytoplankton biomass may be more enhanced in cyclones than in anticyclones. The results agree with the negative correlation between SSH and $[CHL]_{surf}$ anomaly found by Gaube et al., (2014) within the GOM suggesting higher [CHL] concentrations in cyclones.

### 3.3 Underlying processes: nutrient supply to the surface layer

### 3.3.1 Estimations of nutrient concentrations

To better understand the possible mechanisms that explain the differences observed in the [CHL] field within the seasons and the two eddy groups, we address here the role of nutrients (here nitrate + nitrite, [NN]). In the absence of direct measurements, the vertical distribution of [NN] along float trajectories was estimated using XIXIMI-2 and XIXIMI-3 data. The objective was to infer the [NN] from float density profiles. Indeed, when [NN] are plotted against density (Fig. 6), three different layers can be distinguished: the surface layer where [NN] are exhausted; the intermediate layer where [NN] almost linearly increases

with density, and the deep layer within which [NN] are decreasing. In the intermediate layer, which corresponds roughly to the pycnocline and the nitracline (black points on Fig. 6), we estimated the nitracline depth ($Z_N$) and the nitracline steepness ($S_N$) from linear regression ($R^2 = 0.91$). Upper and lower limits of the intermediate layer (respectively 1025.5 kg m$^{-3}$ and 1027.4 kg m$^{-3}$) were chosen according to density criteria (since our goal was to infer the [NN] from float density profiles), and in order to minimize the error of the linear regression of [NN] versus density. In this way, the intermediate layer extends from

the [NN] depleted layer to the [NN] maximum.

The linear fit to the [NN] versus density data (red line in fig. 6) is:

$$[NN] = 16.58(\pm 0.59)\, \sigma_\theta - 422.93(\pm 15.69) \tag{2}$$

where $\sigma_\theta$ is the potential density anomaly (the numbers in parenthesis are the 95% confident intervals). According to Omand and Mahadevan (2015), we can then find the [NN] depletion density $\sigma_\theta(0)$, where [NN] goes to zero as: $\sigma_\theta(0) = 422.93/16.58 = 25.5$ kg m$^{-3}$. $\sigma_\theta(0)$ represents the deepest isopycnal at which nitrate + nitrite is depleted (named also the nitrate depletion



density, Kamykowski and Zentara, 1986). As a comparison, $Z_N$ was compared to the nitracline depth estimated by Jolliff et al., (2008) using the 23.2°C-isotherm with good agreement between the two methods ($R^2 = 0.96$). $S_N$ was also estimated, and can be deduced from Eq. 2:

$$S_N = \frac{\Delta[NO_3]}{\Delta Z} = \frac{16.58.(\sigma_{Z2} - \sigma_{Z1})}{Z2 - Z1} \tag{3}$$

Thus by choosing $\sigma_{Z2}$ and $\sigma_{Z1}$ as the lower and upper limits of the intermediate layer (*i.e.* 1025.5 kg m$^{-3}$ and 1027.4 kg m$^{-3}$), $S_N$ can be determined from float density profiles (Fig. 7).

### 3.3.2 Estimations of nutrient concentrations

In the deepwater GOM, the deepening of the mixed layer in winter is assumed to carry cold and nutrient-rich subsurface water into the euphotic zone, in agreement with the annual cycle of the satellite surface chlorophyll (Jolliff et al., 2008; Müller-Karger et al., 1991). However, our analysis of the [CHL] over the whole water column suggests that the winter [CHL]$_{surf}$ increase does not reflect a real increase in phytoplankton biomass resulting from new nutrient availability. This hypothesis is now tested by considering estimated [NN] along float transects. Fig. 7, which represents the monthly mean and standard

deviation of the nitracline depth and the nitracline steepness, shows that $Z_N$ is always found at depth (regardless of the group) and never approaches the surface, even in winter. This result means that there is no NN accumulation in surface waters and that the deep nutrient reservoir is always isolated from the surface layer (or that nutrient refueling is very small and slower than its uptake by biota). Indeed, the MLD is generally not deep enough to exceed $Z_N$. In our dataset, a MLD much deeper than the inferred $Z_N$ was observed only once (in AG), January 23$^{th}$, 2014. During this event, the MLD reached 171 m (Fig. 4,

the maximum value measured by the floats), and the [CHL]$_{tot}$ reached more than 60 mg m$^{-3}$, *i.e.* twice the mean winter [CHL]$_{tot}$ value (*i.e.* 0.22 mg m$^{-3}$). Apart from this event, it is likely that on average, there are no significant inputs of nutrients by vertical mixing to sustain significant winter new primary production (NPP). This is in full agreement with results obtained from [CHL]. Thus, the idea that winter production in the GOM is enhanced in winter by new nutrients availability may be a misconception. That is at least what it is demonstrated by the float measurements and nutrient estimations. Note however that our results are

limited by the temporal resolution of the floats' profiles (*i.e.* 14 days). This is particularly critical in winter, when the question of the biomass response to MLD deepening events is addressed. The variability in MLD and [CHL]$_{tot}$ (and also in $Z_N$) deduced from bi-monthly profiles is likely underestimated, because mixing events are shorter than the temporal interval of the measurements. Our dataset only demonstrates that a [CHL]$_{tot}$ increase in winter could be exclusively observed in specific areas and/or episodically (*i.e.* when the MLD is very deep and reaches the nitracline), and that such events do not contribute

noticeably to the climatology.





### 3.3.3 Nutrient vertical distribution in cyclones and anticyclones

Float data also showed that the mesoscale activity is a main source of variability for the [CHL] field in the deepwater GOM. In particular, a higher phytoplankton concentration was measured in cyclones with respect to anticyclones (Fig. 5). Fig. 7 indicates that the [NN] distribution is also potentially modulated by the presence of mesoscale structures. Thus, $Z_N$ is

significantly shallower in cyclonic than in anticyclonic structures (p < 0.05), around 80 m in CG and 140 m in AG. This result is consistent with a shallower and intensified DCM in CG than in AG, and in agreement with the conventional view, namely an upward doming of isopycnal surfaces accompanied by a shallowing of the nutricline and an elevated biomass in cyclones (McGillicuddy and Robinson, 1997; McGillicuddy et al., 1998; Oschlies and Garcon, 1998). However, understanding the factors that favor and maintain an enhanced biomass in cyclones is still debated, and the literature addresses a range of

processes (see McGillicuddy (2016) for a review). Our approach allows us to explore at least one mechanism: the role of the vertical flux of [NN] from below, *via* vertical diffusion. This flux is generally considered proportional to the nitrate vertical gradient ($S_N$) through the relationship $F_N = K_z . S_N$ (with $Kz$ the diffusion coefficient, Okubo, 1971). The latest estimate of $K_z$ for the interior Gulf of Mexico is $0.15 \ 10^{-4} \ \text{m}^2 \ \text{s}^{-1}$ (Ledwell et al., 2016), which is similar to what is observed in the open ocean (e.g. Ledwell et al., 1998). Thus considering $K_z$ constant, a steeper nitracline in CG (p < 0.05, Fig. 7) suggests a higher upward

diffusive flux in cyclones with respect to anticyclones. Mean $F_N$ in CG and AG were estimated to be around 26 and 23 mmol $\text{m}^{-2} \ \text{yr}^{-1}$, respectively. As a consequence, the NPP (based on the vertical diffusive flux of NN through the pycnocline) would also be higher in CG than in the AG. This higher NPP could thus be a factor contributing to the observed enhanced biomass in cyclones, as already suggested by previous studies (Biggs et al., 1988; Biggs, 1992; Biggs and Müller-Karger, 1994; Yoder and Mahood, 1983; Zimmerman and Biggs, 1999), although it is difficult to assess with our database. Regenerated production,

local regeneration (Belabbassi et al., 2005) and grazing (Banse, 1995) could also have a fundamental influence, but the answer to this question requires other measurements that are not at our disposal (e.g. oxygen). Nevertheless, we can note that the estimated NPP in CG is higher than in AG by a factor $1.13 \pm 0.02$, on average, which is surprisingly close to the mean [CHL]$_{tot}$ ratio between CG and AG ($1.15 \pm 0.08$).

### 4 Summary and conclusions

The use of profiling floats equipped with biogeochemical sensors provide continuous vertical profile data over wide areas that cannot be obtained otherwise at reasonable cost. The recent deployment of such platforms in the Gulf of Mexico (GOM) generated a remarkable and unique dataset which covers a five-year period. It allowed us to study the variability in phytoplankton biomass using *in situ* data across the region, at a spatio-temporal resolution not reported before. Measurements provided information about the seasonal cycle at the surface and at depth, allowing to study the influence of physical processes

on the deep chlorophyll maximum (DCM), and the identification of the Gulf of Mexico (GOM) as an oligotrophic system. The main findings are:





(1) The surface chlorophyll ([CHL]$_{surf}$) annual pattern viewed by satellite is confirmed, and mixed layer depth (MLD) dynamics appears to be the main factor controlling this cycle.

(2) When considering the integral content of chlorophyll ([CHL]$_{tot}$), no seasonal variability is observed.

(3) [CHL]$_{tot}$ combined with the analysis of backscattering (bbp) data suggest that the total phytoplankton biomass is relatively constant at an annual scale, and that the winter increase in [CHL]$_{surf}$ is primarily associated with a vertical redistribution of chlorophyll or photoacclimation processes, rather than a true biomass increase.

(4) In addition, our observations suggest that the winter mixed layer generally does not reach sufficiently deep to provide large quantities of nutrients to the surface (although some episodic events of [CHL]$_{tot}$ increase associated to very deep mixed layers produced by winter storms cannot be discarded). This result stands in contradiction with the current paradigm of an enhanced primary production in winter, triggered by nutrient input through vertical mixing (Jolliff et al., 2008; Melo Gonzalez et al., 2000; Müller-Karger et al., 1991; Müller-Karger et al., 2015; Salmerón-García et al., 2011).

(5) Float profiles also reveal the subsurface CHL dynamics which cannot be viewed by satellite observations. The temporal variability of the deep chlorophyll maximum (DCM) appears coherent with isopycnal vertical excursions, and a shallower and intensified DCM is found in cyclonic-like structures, *i.e.* when isopycnals are uplifted.

(6) The subsurface [CHL] increase in cyclones is also accompanied by a noticeable bbp increase, supporting that phytoplankton biomass is higher than in anticyclones. A potential but not conclusive explanation is a higher nutrient diffusive flux in cyclones that could contribute to strengthen the new primary production. This suggests that, at the annual scale, the impact of mesoscale features on the phytoplankton biomass may be more important than seasonal processes.

(7) This analysis mainly considered the ecosystem from a "bottom-up" perspective, and we focus mostly on resources regulating phytoplankton growth (light and nutrients) rather than factors influencing losses (grazing, mortality). Other processes, such as submesoscale features (Klein and Lapeyre, 2009) or river run-off (Lohrenz et al. 1997) were not addressed in this study, although they could potentially impact the [CHL] distribution, particularly at shorter spatio-timescales than the ones analyzed in this study (Johens and DiMarco, 2008).

(8) Further deployments of bio-optical profiling floats in the GOM equipped with other biogeochemical sensors, such as nitrate (Johnson and Coletti, 2002; Pasqueron de Fommervault et al., 2015) or oxygen (Körtzinger et al., 2004; Riser and Johnson, 2008), and an increase in the temporal resolution of the profiles would significantly improve our understanding of the mechanisms controlling primary production in the GOM.

(9) Another realistic alternative lies on the use of coupled biochemical/physical models to take advantage of the comprehensive 4D vision they provides in terms of physics, nutrient dynamics and simulated biological processes. However, at this time, the realism of numerical tools still needs to be improved in the GOM. The major barrier to this has been the lack of *in situ* observations over the water column (specifically in deep waters) which remain essential for model validation (Walsh et al., 1989). We are currently using this valuable dataset to calibrate a coupled biochemical/physical model (NEMO-PISCES) and evaluate its performances in the GOM with the objective to address processes that give rise to the results observed in this study (Damien et al., *in prep.*).





**Acknowledgements**

The APEX floats were part of the ''Lagrangian Study of the Deep Circulation in the Gulf of Mexico'', funded by the Bureau of Ocean and Energy Management, USA. Data acquisition and preprocessing thanks to H. Furey, T McKee, and A. Ramsey (Woods Hole Oceanographic Institution). O. Pasqueron de Fommervault acknowledges a postdoctoral scholarship from CICESE, funded via a grant of the National Council of Science and Technology of Mexico - Secretariat of Energy - Hidrocarbons Trust, project 201441. This is a contribution of the Gulf of Mexico Research Consortium (CIGoM). The authors are grateful to Victor Camacho-Ibar, Instituto de Investigaciones Oceanológicas, UABC-Ensenada, who provided nutrient data measured during the XIXIMI-2 and XIXIMI-3 cruises (funded by INECC-SEMARNAT, FOINS-CONACYT, Mexico).

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



**Table 1: Practical information about float Mission and $\alpha$ and $\beta$ coefficients values used for the calibration of fluorescence profiles measured by apex floats.**

| Float | 4902284_ G4901600 | 4902285_ G4901601 | 4902286_ G4901602 | 4902289_ G4901603 | 4902287_ G4901604 | 4902290_ G4901606 |
|---|---|---|---|---|---|---|
| start date | 20 Jul 2011 | 12 Jan 2012 | 16 Jan 2012 | 24 Jun 2012 | 11 Jul 2012 | 25 Sep 2012 |
| end date | 18 Aug 2013 | 11 Nov 2015 | 18 Nov 2015 | 19 Nov 2015 | 21 Nov 2015 | 12 Nov 2015 |
| number of profiles | 83 | 99 | 99 | 88 | 87 | 81 |
| $\alpha$ | 0.63 | 0.63 | 0.66 | 0.59 | 0.60 | 0.66 |
| $\beta$ | 0.030 | 0.050 | 0.039 | 0.026 | 0.031 | 0.028 |

**Table 2: Seasonal mean and standard deviation of the phytoplankton maximum (DCM$_{max}$, in mg m$^{-3}$) and the depth of the phytoplankton maximum (DCM$_Z$, in m). Values were obtained by considering only profiles for which the MLD is shallower than the DCM.**

| | | Dec-Feb | Mar-May | Jun-Aug | Sep-Nov |
|---|---|---|---|---|---|
| All | DCM$_{max}$ | 0.54±0.13 | 0.74±0.23 | 0.64±0.25 | 0.62±0.27 |
| | DCM$_Z$ | 69±17 | 75±19 | 91±21 | 82±24 |
| Cyclonic | DCM$_{max}$ | 0.53±0.14 | 0.80±0.23 | 0.73±0.28 | 0.70±0.29 |
| | DCM$_Z$ | 68±19 | 69±17 | 82±19 | 74±20 |
| Anticyclonic | DCM$_{max}$ | 0.55±0.08 | 0.60±0.18 | 0.53±0.14 | 0.45±0.12 |
| | DCM$_Z$ | 75±11 | 91±15 | 105±17 | 100±22 |

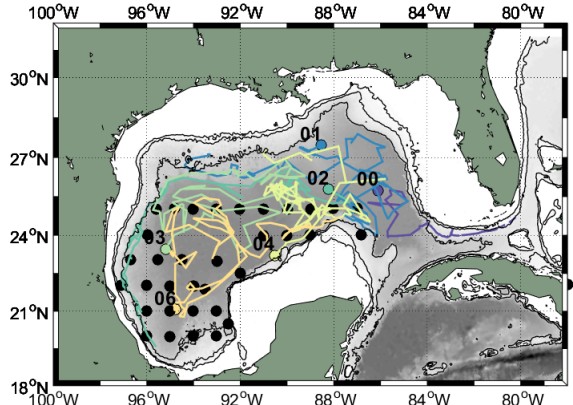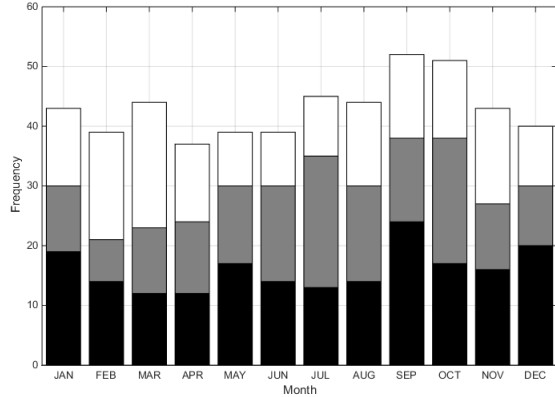

Figure 1: left) Map of the study area, with tracks of BOEM floats (colored lines), and XIXIMI-2 and XIXIMI -3 sampling stations (black circles). Floats position at deployment are indicated by colored circle and numbers. right) Temporal distribution of the profiles acquired by the BOEM (in black cyclonic group profiles, in grey anticyclonic group profiles and in white remaining profiles).





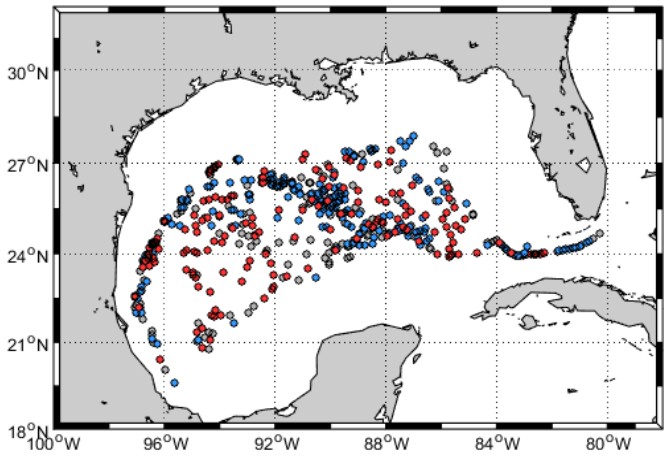

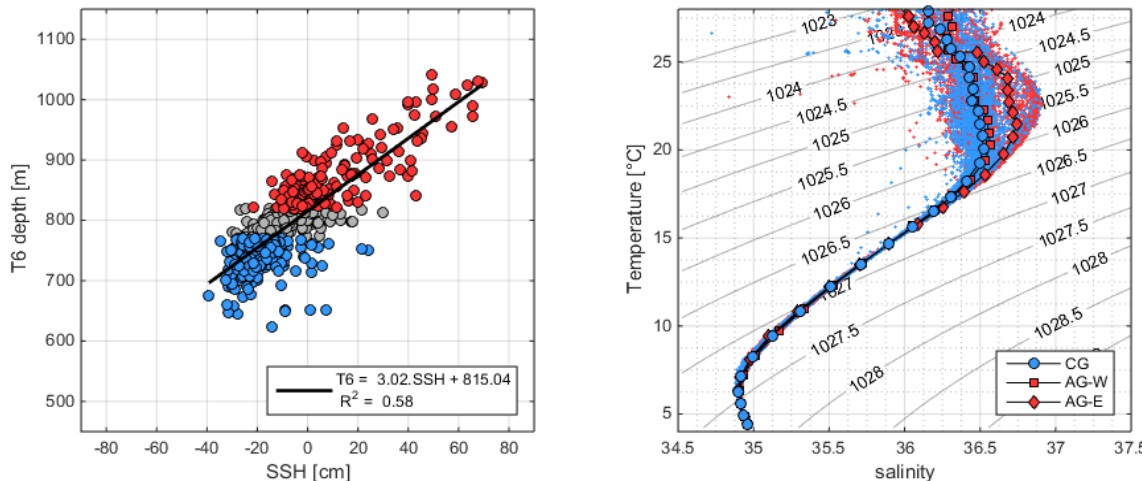

**Figure 2: Top) Position of float profiles. Depth of the 6°C-isotherm derived cyclones and anticyclones are given by blue and red circles. Bottom left) Depth of the 6°C-isotherm versus SSH (gridded data with a spatial resolution of 1/8°, produced by Ssalto/Duacs and distributed by AVISO, with support from CNES (http://www.aviso.altimetry.fr/duacs). To remove seasonal steric effects of large-scale heating and cooling of the upper water column, daily mean SSH calculated in the Gulf of Mexico were systematically subtracted from SSH values. Blue points correspond to the cyclonic group, and red points to the anticyclonic group. Bottom right) T/S diagram. Cyclones are identified in blue (blue circles are mean values) and anticyclones in red (red square are profiles acquired at a latitude west of -88° (west GOM) and red diamond are profiles acquired at a latitude east of -88° (east GOM).**





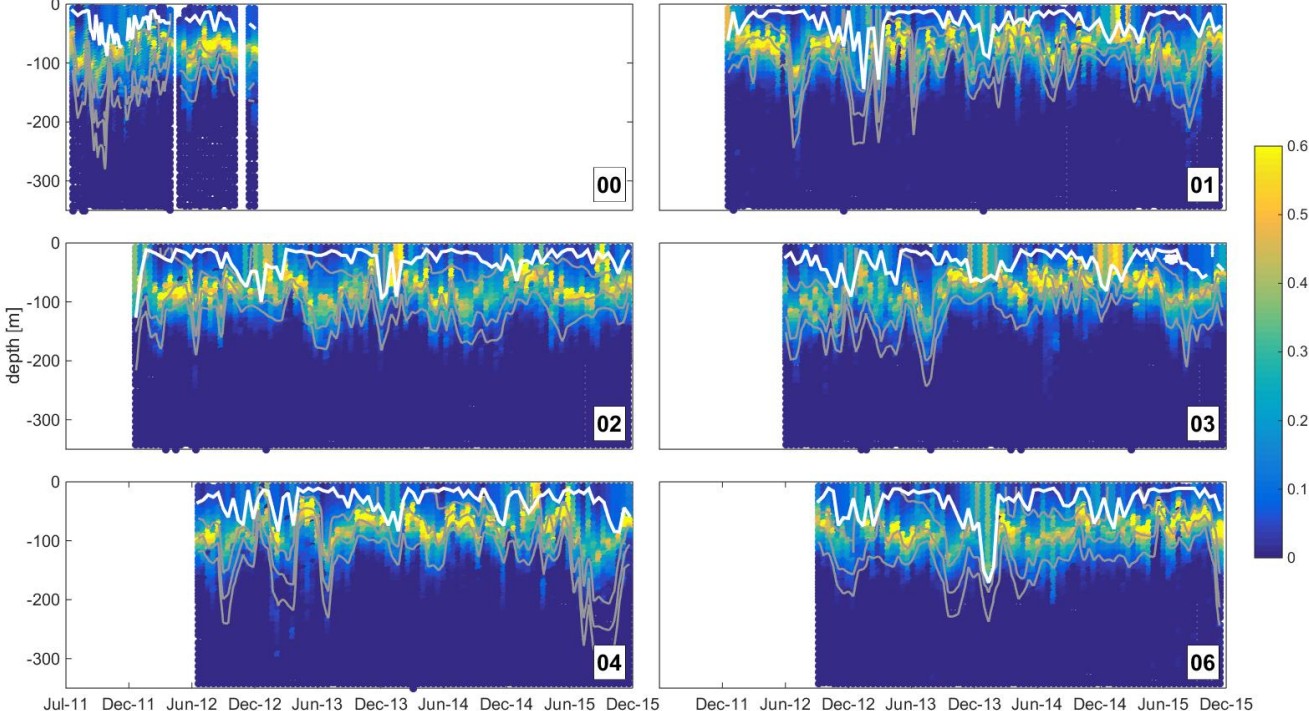

**Figure 3: Calibrated [CHL] float transects in mg m-3. Contour plots of density (1024.5 1025.5 1026 kg m-3, grey lines), and mixed layer depth calculated as the depth where the difference of density from the surface reference, fixed at 10 m-depth, is 0.03 kg m-3 (solid white line), are superimposed.**





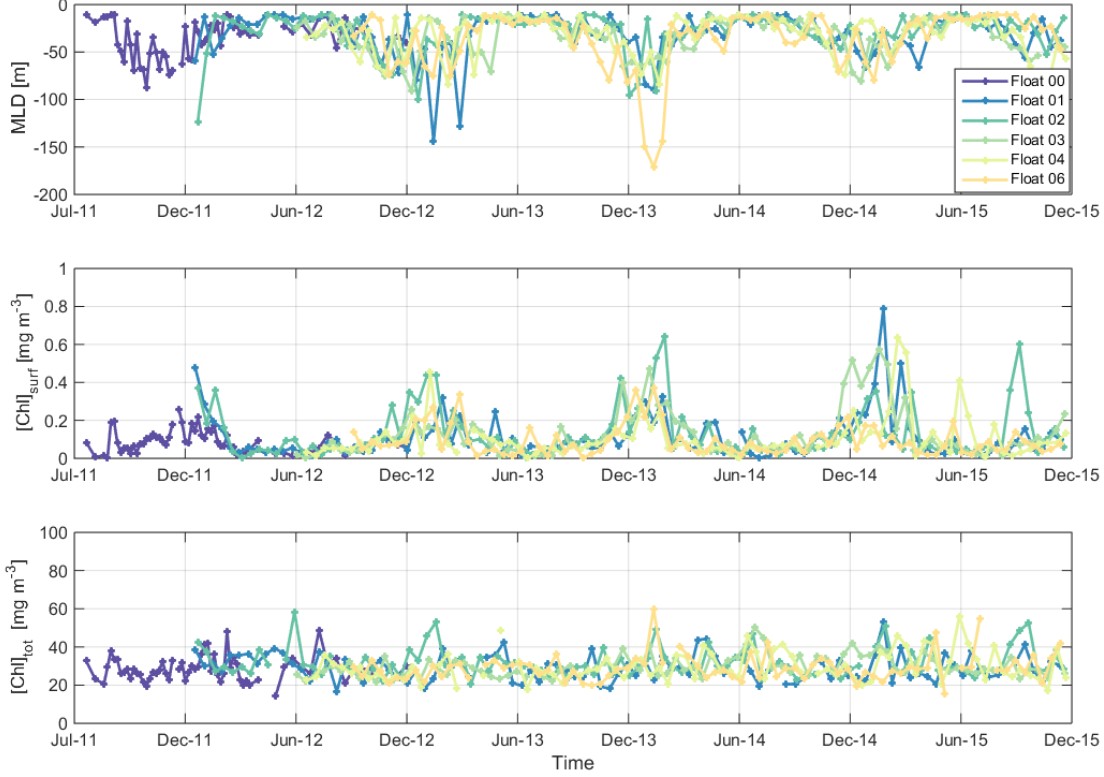

**Figure 4:** Float time-series of the (top) mixed layer depth, (middle) mean surface chlorophyll concentration and (bottom) integrated content of chlorophyll over the 0-350 m layer.





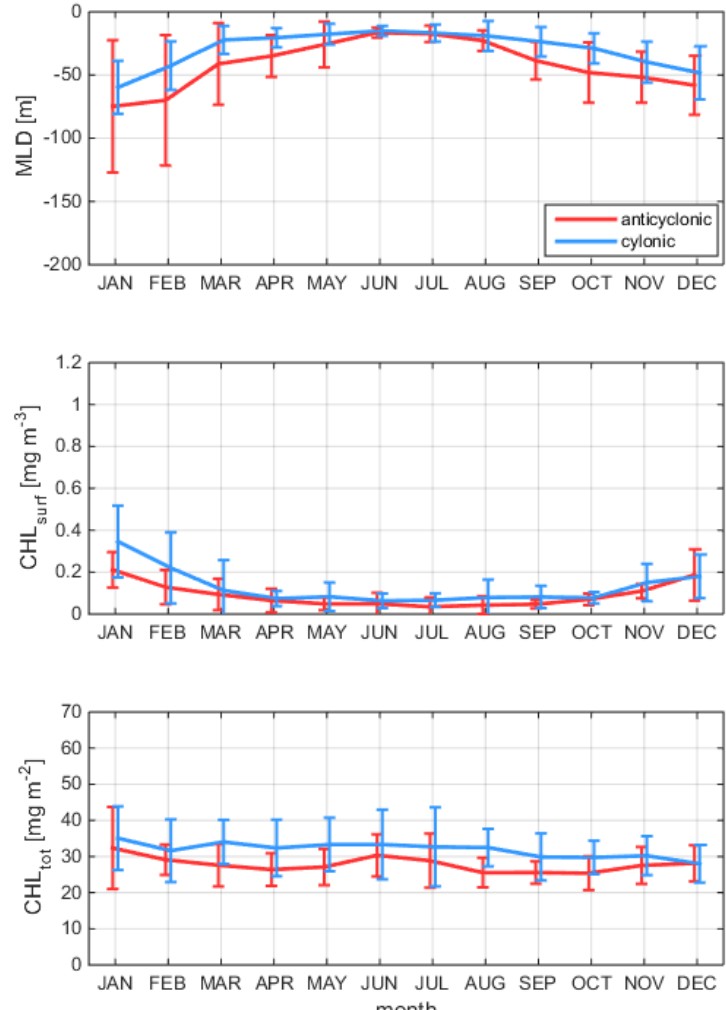

**Figure 5: Monthly mean and standard deviation of: (top) the mixed layer depth, (middle) surface chlorophyll and (bottom) integrated chlorophyll over the first 350m.**





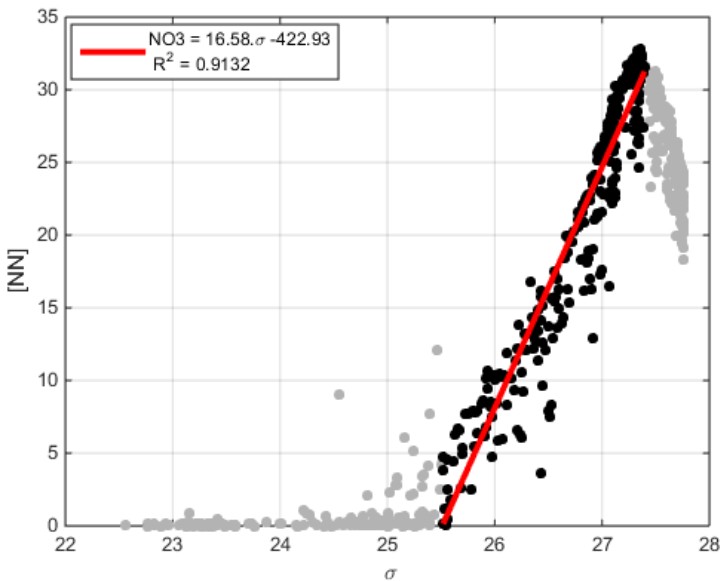

**Figure 6: Nitrate + nitrite concentrations versus potential density anomaly data from XIXIMI-2 and XIXIMI-3 survey cruises. Measurements corresponding to the intermediate layer are plotted in black and were used for the determination of the linear fit (see text).**

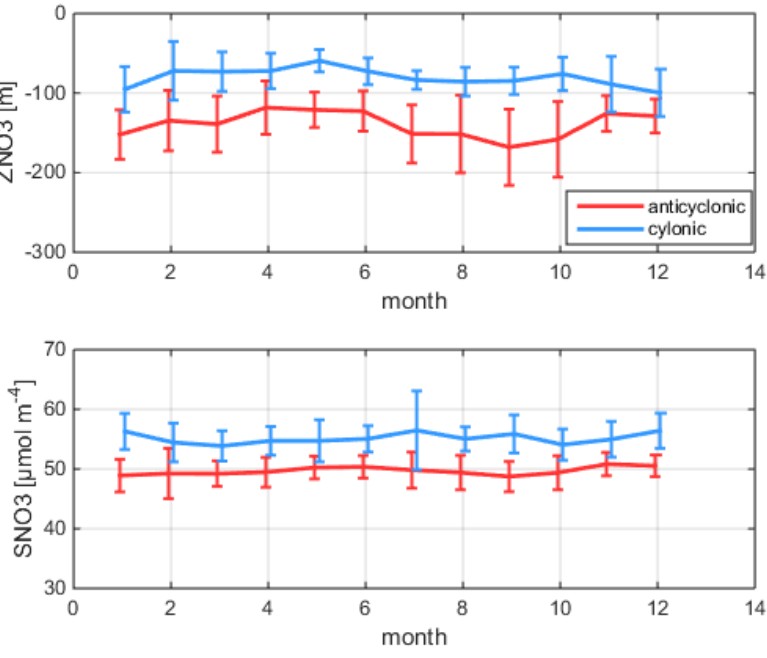

**Figure 7: Monthly mean and standard deviation of: (top) the nitracline depth, (bottom) the nitracline steepness.**