# Peer review of "Temporal variability of chlorophyll distribution in the Gulf of Mexico: bio-optical data from profiling floats"

_Biogeosciences, 2017_

## Referee Comment (RC1) · Anonymous Referee #1 · 26 Jun 2017

This manuscript examines estimates of the temporal and spatial distribution of chlorophyll concentration estimates in the Gulf of Mexico derived from eight profiling floats. These floats provided chlorophyll estimates by in vivo fluorescence. The authors have therefore collected a novel dataset.

The primary objective result of the paper is that the spatial and temporal patterns of surface chlorophyll concentrations derived from the profiling buoys confirms the temporal and spatial chlorophyll estimates observed by ocean color satellites since the late 1970's, and published on extensively. They confirm the seasonal deepening of the surface mixed layer depth in northern winter, in the interior of the Gulf of Mexico,

and shoaling in summer. This has also been published on extensively. The causes for the seasonal surface increase and decrease in chlorophyll, and the spatial patterns defined by circulation have been also explained extensively in the literature over the past 20+ years.

So, in general, the paper finds similar temporal and spatial patterns (i.e., associated with circulation features) in the Gulf of Mexico as have many other people in the past.

The paper is well written in the sense that it flows well, has good prose, and is written in good English.

Yet there are several problems with the paper, which may be serious enough to warrant a very deep revision and withholding publication.

For example, I don't understand what the authors did to compute near-surface chlorophyll concentration from the float data. They say that they took the fluorescence profile, found the highest FLUO value found above 0.9 times the mixed layer depth (MLD) and extrapolated this to the surface (as per Xing et al., 2012). They calibrate this against an ocean color satellite-derived estimate of chlorophyll concentration multiplied times 1.5 the estimated euphotic depth. One problem with this approach is that they make the same assumption of Xing et al (who did a study in the Southern Ocean) that the vertical profile of chlorophyll observed is largely due to quenching of fluorescence, and that the DCM is therefore not 'real'.

The authors probably know that there are data collected and published since the 1960's-1970's to show that the DCM in the Gulf of Mexico is real and seasonal. I wonder if the XIXIMI-2 (July 2011) and XIXIMI-3 (February-March 2013) cruises used by the authors to obtain more than 900 water samples from 74 profiles also had some chlorophyll data?

There are DNA profiles, bacterial profiles, and actual spectrophotometric and HPLC observations that show that the DCA is real and not simply an in vivo chl fluorescence

quenching artifact, as the authors observed. It is not clear to me whether the constant CHLtot seasonal cycle that they find is an artifact of the way they computed the vertical profile with the quenching correction.

It seems a major flaw in this paper is the conclusion that: "the present dataset reveals a vertically integrated content of chlorophyll which remains constant throughout the year, suggesting that the surface increase results from a vertical redistribution of subsurface chlorophyll or photoacclimation processes, rather than a net increase of primary productivity."

The problem is that the integrated water column productivity of a water column with a DCM is not the same as that same water column under a "spring bloom" condition, when phytoplankton biomass is high throughout the mixed layer. The literature is replete with actual measurements of primary productivity that show this. In my opinion, the ecological and biogeochemical interpretation that biomass is the same as productivity is a fatal flaws for this paper. The authors need to go back and fully investigate what mixing can do to phytoplankton blooming in the ocean. They need to review what chlorophyll represents (a crude index of biomass), what productivity is (a rate), and what other factors may play a role in changing these over time and space. What is amazing is that the authors consider past biological oceanographic studies and conclusions of observations in the Gulf of Mexico to be 'beliefs', and proceed to completely misinterpret the chlorophyll signal they observe. They interpret their observations to mean that there are no water-column integrated changes in chlorophyll AND in primary productivity in the Gulf of Mexico. This is clearly a gross misinterpretation of the crude biomass index data they collected. The authors did not exploit the data to make inferences on primary productivity (e.g. perhaps by looking at hour-to-hour and day-to-day changes in biomass). The authors should note that estimates of primary productivity and of chlorophyll concentration are also out of phase in time in the Gulf of Mexico. This has also been reviewed in the literature.

Another problem is the interpretation of nutrient data. The authors have a rich nutrient

dataset with the density data computed from the buoy profiles and the nutrient data from the XIXIMI-2 (July 2011) and XIXIMI-3 (February-March 2013) cruises. The analysis of the density vs. nutrient data is very nice. The problem starts when the authors interpret the nutrient profiles in a biogeochemical and ecological manner. They assume that simply because we see a winter-time increase in chlorophyll concentration in the mixed layer, there also needs to be a clear, measureable signal in nutrient concentrations. Since they don't see this, they conclude that "there are no significant inputs of nutrients by vertical mixing to sustain significant winter new primary production (NPP)". This is incorrect. Nutrients will not be measurable as they are taken up by the phytoplankton. This has been published over and over in the course of the past half century or longer.

The authors seem to somehow dismiss biological oceanography theory in general, including historical knowledge of patterns of vertical distribution of chlorophyll concentration, how these vary in time, and how all this and oceanographic conditions (both biotic and abiotic) affect primary productivity.

Note: the reference: Heileman, S., and Rabalais, 2009, cited to provide a reference on the productivity of the Gulf of Mexico is not a reference for the characterization of productivity in the Gulf. It does not provide summary data. The authors should cite where the actual productivity data comes from that they use to characterize productivity in the Gulf of Mexico. The authors do this often– they cite relatively recent references (in the decade of the 2000's). When they cite earlier literature, they do this in passing and in a dismissive manner, not fully acknowledging that many of the points treated in this paper has already been discussed and explained previously. The problem is that, in doing this, they miss important background knowledge about the oceanography of the Gulf of Mexico. Also, the authors cite studies by Behrenfeld et al (2005), Mignot et al (2014), etc. as suggesting that all temporal changes in chlorophyll observed by satellite are due to changes in pigment concentration in phytoplankton cells. This may be part of what happens, but it is not an accurate characterization of the changes that

occur in the Gulf that they measured.

---

## Author Comment (AC1) · 8 Jul 2017

Please find below our reply to the questions and comments to our submitted manuscript from reviewer #1. They are presented in a point-by-point manner. The numbering of lines or pages refers to the submitted version of the manuscript.

(Reviewer) This manuscript examines estimates of the temporal and spatial distribution of chlorophyll concentration estimates in the Gulf of Mexico derived from eight profiling floats. These floats provided chlorophyll estimates by in vivo fluorescence. The authors have therefore collected a novel dataset. The primary objective result of the paper is that the spatial and temporal patterns of surface chlorophyll concentrations de-

rived from the profiling buoys confirms the temporal and spatial chlorophyll estimates observed by ocean color satellites since the late 1970's, and published on extensively. They confirm the seasonal deepening of the surface mixed layer depth in northern winter, in the interior of the Gulf of Mexico, and shoaling in summer. This has also been published on extensively. The causes for the seasonal surface increase and decrease in chlorophyll, and the spatial patterns defined by circulation have been also explained extensively in the literature over the past 20+ years. So, in general, the paper finds similar temporal and spatial patterns (i.e., associated with circulation features) in the Gulf of Mexico as have many other people in the past. The paper is well written in the sense that it flows well, has good prose, and is written in good English. Yet there are several problems with the paper, which may be serious enough to warrant a very deep revision and withholding publication.

(Authors' response) We thank Referee #1 for his/her quick review and comments. Our intention in this manuscript was to address the basin-wide seasonal average of chlorophyll concentration in the water column (not only at the surface), which is, to our knowledge, not well documented in the Gulf of Mexico. We agree that this data set confirms some of the main aspects of what is known about the seasonal variability of surface chlorophyll. Our main -and we think- new contributions are a) to get time-series of full-water column chlorophyll vertical distribution measurements and b) to make the analysis of the seasonal variability at depth and its relation with the surface chlorophyll content. The reviewer suggests there are some serious problems in our paper. We address all the reviewer comments and criticisms in detail below, hoping they will answer the reviewer concerns and provide a better interpretation of our results.

(Reviewer) For example, I don't understand what the authors did to compute near-surface chlorophyll concentration from the float data. They say that they took the fluorescence profile, found the highest FLUO value found above 0.9 times the mixed layer depth (MLD) and extrapolated this to the surface (as per Xing et al., 2012). They calibrate this against an ocean color satellite-derived estimate of chlorophyll concentration

multiplied times 1.5 the estimated euphotic depth.

(Authors' response) Indeed, the first step of the procedure is to correct profiles for non-photochemical quenching (NPQ). We applied the method of Xing et al. (2012) which is actually implemented in the international BGC-Argo program and consists of extrapolating the highest fluorescence value encountered within the mixed layer up to the surface. Once the fluorescence profile is corrected from the NPQ, we determine instrumental gain and offset using ocean color satellite-derived estimates of chlorophyll concentration. For the comparison against satellite-derived estimate of chlorophyll concentration, the whole 1.5 euphotic layer is used instead of only surface records to minimize the error that would be induced by a wrong NPQ parameterization. Note that the whole procedure is described in detail in Lavigne et al., (2012) paper. So, our near-surface chlorophyll concentration estimates are based on currently accepted international standard procedures (see answer below too).

(Reviewer) One problem with this approach is that they make the same assumption of Xing et al (who did a study in the Southern Ocean) that the vertical profile of chlorophyll observed is largely due to quenching of fluorescence, and that the deep chlorophyll maximum (DCM) is therefore not 'real'. The authors probably know that there are data collected and published since the 1960's-1970's to show that the DCM in the Gulf of Mexico is real and seasonal. I wonder if the XIXIMI-2 (July 2011) and XIXIMI-3 (February-March 2013) cruises used by the authors to obtain more than 900 water samples from 74 profiles also had some chlorophyll data? There are DNA profiles, bacterial profiles, and actual spectrophotometric and HPLC observations that show that the DCA is real and not simply an in vivo chlorophyll fluorescence quenching artifact, as the authors observed.

(Authors' response) We are aware that the calibration/interpretation of fluorescence measurements is critical, and this is why we paid a great deal of attention to the calibration of the data in the manuscript (section 2.2). The Xing et al. (2012) procedure has been validated and applied in various regions (e.g. BATS, HOT, DYFAMED) where

a DCM is present. The relevance of the NPQ correction in these conditions/regions was specifically assessed in Lavigne et al (2012) which shows it has a positive and significant impact on the estimates of chlorophyll. That is the reason why we chose the Xing et al (2012) method for our study.

We would like to point out that we do not say that the DCM is not real or does not exist in the Gulf of Mexico. Quite the opposite, the manuscript shows how this DCM varies in concentration and depth both for the entire time series (Fig. 3) as well as seasonally in the climatological averages shown in Table 2. In fact, the NPQ correction only concerns the mixed layer and as a consequence, the impact of the correction on the fluorescence profile is generally limited to the surface and does not impact the observed DCM (except on the occasions when the mixed layer is deep enough to reach the DCM. Note that during XIXIMI 2 and XIXIMI 3 cruises no HPLC observations were made.

(Reviewer) It is not clear to me whether the constant CHLtot seasonal cycle that they find is an artifact of the way they computed the vertical profile with the quenching correction.

(Authors' response) The vertically integrated chlorophyll ([CHL]tot) depends largely on the chlorophyll content at the DCM. Hence, the contribution of the NPQ correction (which is limited to the mixed layer) is small. This can be verified in Fig. 1 below, which shows times-series from one float (float number 02 in the submitted manuscript) with and without the NPQ correction. One can see the constant [CHL]tot seasonal cycle is not an artifact of the NPQ correction.

(Reviewer) It seems a major flaw in this paper is the conclusion that: "the present dataset reveals a vertically integrated content of chlorophyll which remains constant throughout the year, suggesting that the surface increase results from a vertical re-distribution of subsurface chlorophyll or photoacclimation processes, rather than a net increase of primary productivity."

(Authors' response) The reviewer is absolutely right. The sentence in the abstract (page.1 line 19) "the present dataset reveals a vertically integrated content of chlorophyll which remains constant throughout the year, suggesting that the surface increase results from a vertical redistribution of subsurface chlorophyll or photoacclimation processes, rather than a net increase of primary productivity" is wrong. We thank the reviewer for noticing that, and the term primary production will be changed to biomass, which is what we actually meant and missed to correct before submitting. Note that this was correctly stated in our conclusion number 3 (page 12, lines 4-6).

(Reviewer) The problem is that the integrated water column productivity of a water column with a DCM is not the same as that same water column under a "spring bloom" condition, when phytoplankton biomass is high throughout the mixed layer. The literature is replete with actual measurements of primary productivity that show this. In my opinion, the ecological and biogeochemical interpretation that biomass is the same as productivity is a fatal flaws for this paper. The authors need to go back and fully investigate what mixing can do to phytoplankton blooming in the ocean. They need to review what chlorophyll represents (a crude index of biomass), what productivity is (a rate), and what other factors may play a role in changing these over time and space.

(Authors' response) We fully agree with the reviewer that biomass is not the same as productivity and regret the confusion that our mistake in the abstract caused. Indeed, chlorophyll measurements were described and interpreted in terms of biomass (see section 3.1 and 3.2) whereas primary production (and more precisely new primary production), was evoked when nutrients fluxes are estimated (section 3.3), with the aim to discuss it as a hypothesis and/or a possible mechanism (not as a direct result of chlorophyll measurements). In order to make this clearer, some changes will be made in the manuscript.

The conclusion number 4 (page 12, lines 7-11) will be rewritten:

"In addition, our observations show that the winter mixed layer is generally not deep

enough to reach the nitracline. This suggest that, on average, only small amount of nutrients are potentially injected to the surface layer through vertical mixing, although some short time-scales events of important nutrient inputs associated to very deep mixed layers during winter storms cannot be discarded. This also suggests that nutrients supply by winter mixing is not necessarily the main cause of the seasonal surface chlorophyll variability, although it is difficult to say with certainty with our dataset". instead of: "In addition, our observations suggest that the winter mixed layer generally does not reach sufficiently deep to provide large quantities of nutrients to the surface (although some episodic events of [CHL]tot increase associated to very deep mixed layers produced by winter storms cannot be discarded). This result stands in contradiction with the current paradigm of an enhanced primary production in winter, triggered by nutrient input through vertical mixing."

Sentences in the section 3.3.2 (page 10 lines 21-23) "This is in full agreement with results obtained from [CHL]. Thus, the idea that winter production in the GOM is enhanced in winter by new nutrients availability may be a misconception." will be removed.

For the same reasons and given it implies of lot of suppositions and that it does not add significant information we will remove the sentence in the section 3.3.3 (page 11, lines 21-23): "Nevertheless, we can note that the estimated NPP in CG is higher than in AG by a factor $1.13 \pm 0.02$, on average, which is surprisingly close to the mean [CHL]tot ratio between CG and AG ($1.15 \pm 0.08$)."

(Reviewer) What is amazing is that the authors consider past biological oceanographic studies and conclusions of observations in the Gulf of Mexico to be 'beliefs', and proceed to completely misinterpret the chlorophyll signal they observe.

(Authors' response) The term "belief" used in the manuscript did not mean to disregard nor minimize previous studies at all. We agree term is not appropriate and we have changed "belief" to "interpretation" in the two sentences where it was used (page 1, line 16 in the abstract; page 2, line 8 in the introduction). We were trying to emphasize the

fact that, in the Gulf of Mexico, the seasonal cycle of chlorophyll, at a basin scale, has been almost exclusively addressed using satellite measurements (see the review of Biggs and Ressler, 2001), which only provide surface information. To our knowledge, and prior to the deployment of BOEM floats, available chlorophyll vertical profiles in the Gulf of Mexico did not have the required spatio-temporal resolution to resolve the seasonal cycle of the DCM or the chlorophyll content within the water column at the basin scale.

(Reviewer) They interpret their observations to mean that there are no water-column integrated changes in chlorophyll AND in primary productivity in the Gulf of Mexico. This is clearly a gross misinterpretation of the crude biomass index data they collected. The authors did not exploit the data to make inferences on primary productivity (e.g. perhaps by looking at hour-to-hour and day-to-day changes in biomass). The authors should note that estimates of primary productivity and of chlorophyll concentration are also out of phase in time in the Gulf of Mexico. This has also been reviewed in the literature.

(Authors' response) As we answered above, we agree that we can only address biomass with chlorophyll data, and modifications have been done to make it clearer in the manuscript (see comment above). We indeed observed that the integrated content of chlorophyll does not show a clear seasonal variability, which we interpret as total biomass remaining constant throughout the year at a monthly timescales (which is consistent with the analysis of the bbp data shown in the supplementary material). In addition, the temporal resolution of our floats measurements (14 days) prevented us to infer hour-to-hour and day-to-day changes in biomass to estimate in an appropriate manner primary production. We are currently addressing this question using a coupled biogeochemical/physical model (NEMO-PISCES) with the objective to check to what extent the hypothesis of the present work is validated.

(Reviewer) Another problem is the interpretation of nutrient data. The authors have a rich nutrient dataset with the density data computed from the buoy profiles and the nutrient data from the XIXIMI-2 (July 2011) and XIXIMI-3 (February-March 2013) cruises. The analysis of the density vs. nutrient data is very nice. The problem starts when the authors interpret the nutrient profiles in a biogeochemical and ecological manner. They assume that simply because we see a winter-time increase in chlorophyll concentration in the mixed layer, there also needs to be a clear, measureable signal in nutrient concentrations. Since they don't see this, they conclude that "there are no significant inputs of nutrients by vertical mixing to sustain significant winter new primary production (NPP)". This is incorrect. Nutrients will not be measurable as they are taken up by the phytoplankton. This has been published over and over in the course of the past half century or longer.

(Authors' response) We agree with the reviewer's comment "Nutrients will not be measurable as they are taken up by the phytoplankton", and it is stated page10, line18. We thus will make the following modifications in section 3.3.2 in the hope this helps to clarify our arguments:

Page 10, line 13 "does not necessarily reflect" instead of "does not reflect".

Page 10 lines 14-22 (from "Fig. 7" to "(NPP)") is rewritten as fallow"Fig. 7, which represents the monthly mean and standard deviation of the nitracline depth and the nitracline steepness, shows that ZN is always found at depth and does not show a clear seasonal pattern (regardless of the group). When MLD (Fig. 5) and ZN are compared, one can note, that the winter mixed layer is generally shallower than the ZN. Hence if we assume that large inputs of nutrients can only be expected when the MLD reach below the average ZN, it is likely that nutrients injections by vertical mixing, are low, even in winter. In our dataset, a MLD much deeper than the inferred ZN was observed only once (in AG), January 23th, 2014. During this event, the MLD reached 171 m (Fig. 4, the maximum value measured by the floats), and the [CHL]tot reached more than 60 mg m-3, i.e. twice the mean winter [CHL]tot value (i.e. 0.22 mg m-3). Apart from this event, it is likely that surface water are always nutrients depleted. Nutrient may not be measured in surface as they are taken up by phytoplankton. However, the fact that we

do not observe NN accumulation in surface means that nutrient refueling is relatively small or at least slower than its uptake by biota."

Section 3.3.2 (page 10 lines 21-23), see above.

Conclusion number 4 (page 12, lines 7-11), see above.

(Reviewer) The authors seem to somehow dismiss biological oceanography theory in general, including historical knowledge of patterns of vertical distribution of chlorophyll concentration, how these vary in time, and how all this and oceanographic conditions (both biotic and abiotic) affect primary productivity.

(Authors' response) We have not found literature which address the seasonal variability of the vertical distribution of chlorophyll in the Gulf of Mexico. We would appreciate if the reviewer could point us to those historical articles, given that in the 90s Muller-Karger et al., (1991) stated:

"(. . .) in situ oceanographic data set for the Gulf of Mexico is still insufficient to address questions and processes affecting the distribution of biological and chemical properties."

and recently (Muller-Karger et al., 2015):

"The waters of the interior of the Gulf of Mexico seaward of the continental margin continue to be seriously undersampled. (. . .). We were not able to derive a good chlorophyll concentration dataset from historical field observations archived at the NOAA NODC to compare with either CZCS, SeaWiFS, or MODIS chlorophyll estimates. Most samples in the Gulf of Mexico available at the NODC are from the northern and eastern shelf regions, with relatively few samples available from offshore waters. Thus, information derived from remote sensing is essential for characterization of the deep water areas of the Gulf."

In fact, we are not aware of studies showing mixed layer depth climatologies based on direct measurements for the Gulf of Mexico either, since the ones available are indirect

estimates based on models or parametrizations (Mandoza et al., 2005; Muller-Karger et al., 2015; Zavala-Hidalgo et al., 2014). In this paper we specifically focused on the measured MLD and considered the ecosystem from a bottom-up perspective. We agree that biomass is regulated by a wider range of processes (e.g. biotic processes) which are not directly addressed in our manuscript since to answer these questions would require other measurements that were not at our disposal. This is a limitation to our work which we were careful to mention in the manuscript (section 3.3.2 page 11 lines 19-23, section 4 page13 lines 19-34). Hence, we consider that the new measurements of water column chlorophyll and bpp seasonal variability invite to new hypotheses that are worth exploring.

(Reviewer) the reference: Heileman, S., and Rabalais, 2009, cited to provide a reference on the productivity of the Gulf of Mexico is not a reference for the characterization of productivity in the Gulf. It does not provide summary data. The authors should cite where the actual productivity data comes from that they use to characterize productivity in the Gulf of Mexico.

(Authors' response) Heileman and Rabalais, (2009) is only used in the introduction as a general characterization of net primary production in the Gulf of Mexico. We would appreciate if you could provide us a more appropriate reference.

(Reviewer) The authors do this often– they cite relatively recent references (in the decade of the 2000's). When they cite earlier literature, they do this in passing and in a dismissive manner, not fully acknowledging that many of the points treated in this paper has already been discussed and explained previously. The problem is that, in doing this, they miss important background knowledge about the oceanography of the Gulf of Mexico.

(Authors' response) The main objective of our work was to study the chlorophyll variability in the water column at a basin scale and on a seasonal basis. Given the lack of time series of vertical profiles of chlorophyll in the Gulf of Mexico, except for indirect measurements obtained from satellite or models, we would be very grateful if the reviewer could tell us which relevant bibliography we missed that addressed the seasonal cycle of the chlorophyll content at depth and at the basin scale using in situ measurements.

(Reviewer) Also, the authors cite studies by Behrenfeld et al (2005), Mignot et al (2014), etc. as suggesting that all temporal changes in chlorophyll observed by satellite are due to changes in pigment concentration in phytoplankton cells. This may be part of what happens, but it is not an accurate characterization of the changes that occur in the Gulf that they measured.

(Authors' response) We agree that the observed changes may not only be due to pho-toaclimation, and we do not state that this is the only relevant processes involved. In the discussion section 3.1 (page 7, line 27-34 and page 8, line1-9) and conclusion number 7 (page 12, line 19-23), we also mention other possible mechanisms. However, this new dataset suggests that photoacclimation may be relevant and worth exploring. The above references of Behrenfeld et al., (2005), Mignot et al., (2014) are included in the discussion to support this.
* * *
[Figure]

[Figure]

**Fig. 1.**

---

## Referee Comment (RC2) · Anonymous Referee #2 · 11 Jul 2017

The manuscript Temporal variability of chlorophyll distribution in the Gulf of Mexico: bio-optical data from profiling floats by Pasqueron De Fommervault et al. attempts to evaluate the temporal and spatial variability of chlorophyll concentration in the Gulf of Mexico. The study utilises data from eight bio-optical profiling floats. The paper addresses the winter increase in sea surface chlorophyll concentration and the impact of mesoscale eddies on phytoplankton biomass in the Gulf of Mexico. This is done by applying already published methods to a rather new dataset. The problem is that, most of the time, the methods cannot be applied or are not applied correctly. Consequently, most of the results presented in the manuscript are flawed. Additionally, the manuscript is poorly written, the arguments are hard to follow, and often not justified by the appropriate references. The amount of work needed for publication in Biogeosciences is considerably more than for a major revision.

Comment 1: One of the conclusion of the manuscript is that the winter increase in sea surface chlorophyll concentration is due a photoacclimation process or a re-entrainment of phytoplankton cells at depths. I felt like the authors have chosen to cop-out on testing one hypothesis over another. The authors have all the data necessary to investigate what causes the winter increase in sea surface chlorophyll concentration in the Gulf of Mexico. They need to do more than just commenting on the float observations.

Comment 2: The authors use the depth of the 6C isotherm to classify the eddies in the Gulf of Mexico. They argued that this isotherm has a mean depth of 795 m and that Bung et al (2002) found that this isotherm separates the deep stable water from the eddy-influenced surface water. Finally, it is said that Hamilton et al. 2017 found a strong correlation between the isotherm and upper layer eddies. First, Bung et al. (2002) did not identify eddies with this isotherm. It was used to delineate the depth of the Loop Current in the Yucatan channel. Consequently, the use of the 6C isotherm to identify mesoscale eddies cannot be justify by this reference. Second, the vertical extent of eddies core is comprised, in average, between 300 and 400 m. What is the rationale of using such a deep isotherm to detect eddies that impact the first 400m of the water column? Third, the authors claimed that Hamilton et al. (2017) found a correlation between mesoscale eddies and the 6C isotherm. However, this reference seems to be an oral presentation and again it cannot be used to justify the utilisation of the 6C isotherm.

Comment 3: The section on the impact of mesoscale eddies on phytoplankton biomass is largely inspired by the study of Dufois et al. 2014. The authors compared the float observations when the floats were profiling in a cyclonic structure with the observations when the floats were profiling in an anticyclonic structure. By averaging observations that were collected at different times of the year and locations in the Gulf of Mexico,

you are looking at signals that are both influenced by the seasonal and large scale variability and the mesoscale activity; this is not correct. To properly assess the impact of eddy signal on a given variable, one needs to look at the departure from the seasonal mean (see Cushman-Rosin, 1994). All sections about the role of mesoscale structures need to be changed.

Comment 4: The depth of an isopycnal surface cannot be used to determine a nitracline depth. In the open ocean, nutrient concentrations are controlled by both physical processes such as vertical/horizontal advection and diffusion, convection, and biological processes such as phytoplankton growth, remineralization, etc... At a pinch, Eq. (2) can be used to give a crude estimation of the nitracline depth for a quasi-1D steady-state system with a surface layer depleted in nutrients, with no change in solar radiation and mixed layer depth. In your case, these assumptions do not hold. Sections 3.3.3 and 3.3.3 need to be removed and the conclusion need to be changed accordingly.

---

## Author Comment (AC2) · 31 Jul 2017

Please find below our reply to comments to our submitted manuscript from reviewer #2. The numbering of lines or pages refers to the submitted version of the manuscript.

(Reviewer) The manuscript Temporal variability of chlorophyll distribution in the Gulf of Mexico: bio-optical data from profiling floats by Pasqueron De Fommervault et al. attempts to evaluate the temporal and spatial variability of chlorophyll concentration in the Gulf of Mexico. The study utilises data from eight bio-optical profiling floats. The paper addresses the winter increase in sea surface chlorophyll concentration and the impact of mesoscale eddies on phytoplankton biomass in the Gulf of Mexico. This is

done by applying already published methods to a rather new dataset. The problem is that, most of the time, the methods cannot be applied or are not applied correctly. Consequently, most of the results presented in the manuscript are flawed. Additionally, the manuscript is poorly written, the arguments are hard to follow, and often not justified by the appropriate references. The amount of work needed for publication in Biogeosciences is considerably more than for a major revision.

(Authors' response) We thank Referee #2 for his/her review and comments. The major criticisms are related to methodological issues that prevent, according to the reviewer, an accurate analysis of our dataset. Below, we address and discuss all the comments hoping they will clarify our work and answer the reviewers' concerns. Additional references and figures are also presented. The comment that the manuscript is poorly written and the arguments hard to follow is difficult to address since no specifics are provided. We note however, that reviewer #1 considers that the paper is "well written" and that it "flows well" (even if he/she has other concerns that we have addressed in the corresponding reply).

(Reviewer's comment 1) One of the conclusion of the manuscript is that the winter increase in sea surface chlorophyll concentration is due a photoacclimation process or a reentrainment of phytoplankton cells at depths. I felt like the authors have chosen to copout on testing one hypothesis over another. The authors have all the data necessary to investigate what causes the winter increase in sea surface chlorophyll concentration Gulf of Mexico. They need to do more than just commenting on the float observations.

(Authors' response) In the manuscript it is suggested that the winter surface chlorophyll increase could result from (i) photoacclimation within the mixed layer (ML) due to a reduction in average light availability, and/or (ii) from mixing of the deep chlorophyll maximum (DCM) into the ML. Processes involved in (i) are considered on a seasonal basis. The hypothesis is that during the winter mixing period, the average light intensity for phytoplankton is reduced in the ML with respect to summer conditions leading to

an increase of intracellular chlorophyll content. Since, on average, a low bbp/[chl] ratio at the surface is persistent in winter, this hypothesis seems reasonable. Hypothesis (ii) supposes that low-light acclimated cells from the DCM are transported to the surface by mixing process, thus increasing the bbp/[chl] ratio at the surface. Considering only (ii) assumes that phytoplankton cells do not have time to re-acclimate to their new light environment (hours to days processes). So it is not warranted to test (ii), given the temporal resolution of our floats (two profiles per month from each float). Contrary to the reviewer's claim, we do not think the data have enough temporal resolution to determine what causes the winter increase in surface chlorophyll. That is the reason why we are cautious and suggest that is difficult to attribute the observed changes in chlorophyll to only one mechanism and suggest/speculate possible mechanisms. Unless the reviewer can point towards specific references or analysis tools that could help resolve this question with the temporal resolution and parameters measured by the floats, we stand by our original conclusion that this cannot be done with the data available.

(Reviewer's comment 2) The authors use the depth of the 6C isotherm to classify the eddies in the Gulf of Mexico. They argued that this isotherm has a mean depth of 795 m and that Bunge et al (2002) found that this isotherm separates the deep stable water from the eddy-influenced surface water. Finally, it is said that Hamilton et al. 2017 found a strong correlation between the isotherm and upper layer eddies. First, Bunge et al. (2002) did not identify eddies with this isotherm. It was used to delineate the depth of the Loop Current in the Yucatan channel. Consequently, the use of the 6C isotherm to identify mesoscale eddies cannot be justify by this reference. Second, the vertical extent of eddies core is comprised, in average, between 300 and 400 m. What is the rationale of using such a deep isotherm to detect eddies that impact the first 400m of the water column? Third, the authors claimed that Hamilton et al. (2017) found a correlation between mesoscale eddies and the 6C isotherm. However, this reference seems to be an oral presentation and again it cannot be used to justify the utilization of the 6C isotherm.

(Authors' response) First, note that the analysis is meant to detect mesoscale structures (i.e nearly geostrophic flow structures that can be identified from sea surface height anomalies / thermocline depth anomalies) but also structures that are part of the mean circulation. In the Gulf of Mexico (GOM), besides transient eddies such as the anticyclones detached from the Loop Current (LC), the cyclones that often accompany the detachment process and other eddies present inside the GOM, there are semi-permanent structures that, for example, can be seen in the figure below (Figure 1) which shows the mean dynamic topography of the GOM (Rio and Hernandez, 2004, and mean-dynamic topography.html). One can see that the LC itself (anticyclonic), the cyclonic gyre of the Bay of Campeche and the (anticyclonic) eddy/gyre straddling the central GOM (produced in part by the path of LC eddies inside the GOM) are features of the mean circulation.

Hence, to avoid misunderstanding and to make sure it is clear we are referring to mesoscale structures and not only "eddies" which are generally understood as transient and closed circulation features, the term eddy in page 5, line 27 is now replaced by mesoscale, and removed page 9 line 16.

Second, in several references (e.g. Donohue et al., 2007; Hamilton 2007; Sheinbaum et al., 2007; Donohue et al., 2008; Hamilton et al., 2016) it is shown that the vertical displacement of the 6 to 10°C isotherms (located at the base of the LC and LC eddies) is anti-correlated with sea-surface height displacements, highlighting the importance of first baroclinic mode dynamics in the GOM. 6-7°C isotherms are at 450-500 m depth within some cyclonic eddies. Donohue et al., (2008), Cox et al., (2010) and Hamilton et al., (2016) suggest the 6°C isotherm is a better choice for a two-layer analysis, and Kolodziejczyk et al., (2012) find the zero crossing of the first baroclinic mode near 1000 m depth.

We agree with the reviewer that not enough references or explanations for using the 6°C isotherm depth (T6) variability used to identify mesoscale structures were given in the submitted manuscript. Thus, To make a stronger case for the use of T6 to identify

mesoscale structures, new references are added and list items of pages 5 (lines 30-31) and page 6 (lines 1-4) are re-written:

- The GOM can be, to some extent, be studied as a two-layer baroclinic system, with vertical displacements of the 6 to 10°C isotherms (located at the base of the Loop Current and Loop Current eddies and at 400-600 m depth in cyclonic eddies) being anti-correlated with sea-surface height (Donohue et al., 2007; Hamilton 2007; Sheinbaum et al., 2007; Donohue et al., 2008; Hamilton et al., 2016, Hamilton et al., 2017).

- T6 is in the lower thermocline and has been used as the interface for a two layer system, separating deep waters from the more energetic upper layer containing mesoscale structures and the Loop Current (Bunge et al., 2002; Donohue et al. 2008; Hamilton et al. 2016, Hamilton et al., 2017).

Third, the depth of T6 is directly measured by the floats. Hence we decided to use this parameter to track mesoscale structures instead of sea surface height (SSH) measured by satellites, which have limited spatial resolution so the required interpolation to the locations of our profiles would add additional errors (as it is shown in the manuscript, using SSH instead of T6 leads to essentially the same results, figure S1 in supplementary material).

Finally, note that the baroclinic signal of the mesoscale structures reaches down below 600m (or more) in the GOM (e.g. Lewis et al., 1989; Donohue 2016a and b, Sheinbaum et al., 2007, Hamilton et al., 2017), even if the eddies' core is shallower. Additionally, Hamilton et al., (2017) reference given in the manuscript is not an oral presentation (it is a BOEM-MMS technical report which is (now) available online (https://www.boem.gov/ESPIS/5/5583.pdf). The link will be included in the reference list.

Given the above, we decided to leave the analysis as is, using the T6 criteria to obtain the anticyclonic/cyclonic subsets. However, we can certainly change all the analysis using SSH instead, if the reviewer and editor consider that is a better choice. If so, we

do ask for additional time to recalculate this to include the corresponding new figures, table 2 as well as the values and statistical significance analysis discussed in the text.

(Reviewer's comment 3) The section on the impact of mesoscale eddies on phytoplankton biomass is largely inspired by the study of Dufois et al. 2014. The authors compared the float observations when the floats were profiling in a cyclonic structure with the observations when the floats were profiling in an anticyclonic structure. By averaging observations that were collected at different times of the year and locations in the Gulf of Mexico, you are looking at signals that are both influenced by the seasonal and large scale variability and the mesoscale activity; this is not correct. To properly assess the impact of eddy signal on a given variable, one needs to look at the departure from the seasonal mean (see Cushman-Rosin, 1994). All sections about the role of mesoscale structures need to be changed.

(Authors' response) We did not "average observations that were collected at different times of the year and locations in the Gulf of Mexico" as the referee writes, since we did not average different times of the year. We clustered the data in different groups based on identifying whether the profiles were taken in anticyclonic, cyclonic or neutral vorticity anomalies, following the criteria explained in the text (see also comment 2 above), and then simply computed monthly climatologies for the anticyclonic and cyclonic subsets, gathering all profiles corresponding to each month regardless of the year and the geographic position, and looked for differences between these two clusters. Data are evenly distributed so each month of the year has a similar number of profiles (figure 1, right panel in the manuscript). One may question the representativeness of this climatology given the amount of data (two vertical profiles per month during 5 years from seven APEX floats), but it is what is available now and sufficient, in our view, to make this computation worthwhile.

It is well known that the seasonal cycle of geophysical and biogeochemical variables is not monochromatic and has time scales that overlap those of mesoscale turbulence. Both processes are thus impossible to separate properly, "even by defining running

variances over 6-month windows or by subtracting a mean seasonal cycle from the time series before variance computations" (Penduff et al., 2004). So a decomposition of a variable as a time-mean+seasonal+"eddy" is not necessarily warranted as a means to separate seasonal and mesoscale eddy signals. Besides, depending on how calculations are made, the "eddy" part will include lower frequency variations too. Although there is certainly seasonal variability in the GOM related to atmospheric forcing (air-sea interaction river run-off), spectra (e.g. SSH) shows comparable or more energetic variations related to the mesoscale, in a band of frequencies that overlaps with seasonal variations (3-4 months, Jouanno et al, 2016; Hamilton and Lee, 2005).

Having said that, we followed the suggestion of the reviewer to first remove the seasonal signal to try to separate it from the mesoscale variability. We first calculate the climatology for the variable in question using all profiles regardless of vorticity anomaly, and then substract the monthly mean value from each profile to compute the anomalies. These anomalies have now mesoscale as well as lower frequency (interannual) signals. The profiles are then separated in the two clusters (cyclonic and anticyclonic), and a monhtly climatology of the anomalies is obtained. Figures below (Figures 2 and 3), same as figure 5 and figure 7 in the manuscript but for anomalies, show the same results as those obtained from the original calculation. Namely that anticyclones tend to show a deeper mixed layer depth (MLD) than cyclones in winter, that cyclones have slighlty larger integrated chlorophyll content than anticyclones, and that the nitracline depth and steepness is larger for cyclones throughout the year.

We decided not to use the anomalies, given that they are only useful to show relative differences between the two groups (anticyclonic vs cyclonic) of a given parameter, but it becomes complicated when two parameters are to be compared (e.g. the depth of the ML with that of the DCM). We think the discussion is clearer if we stick to the absolute values, and given the results above, we hope that the reviewer agrees with us. However, the title of the section will be changed:

"Impact of mesoscale structures on the annual cycle" Instead of "Role of mesoscale

structures in shaping the annual cycle"

and we will add this paragraph to the corresponding section so as to make a more convincing case for our approach (page 8 line 20):

"It is well known that the seasonal cycle of geophysical and biogeochemical variables is not monochromatic and has time scales that overlap those of mesoscale turbulence (e.g. Penduff et al., 2004) thus, it is not possible to separate them properly. Although there is certainly seasonal variability in the Gulf of Mexico related to atmospheric forcing (air-sea fluxes and river run-off), spectra shows comparable or more energetic variations related to the mesoscale in a band of frequencies that overlaps with seasonal variations (3-4 months for sea surface height, e.g., Jouanno et al, 2016, Hamilton and Lee, 2005). Hence, it could be expected that the variability observed at a given period of the year also depends on the presence of mesoscale structures. We therefore analyzed the seasonal cycle gathering data on a monthly-basis and considering separately profiles acquired in cyclonic and anticylonic structures, to assess the differences between the two groups."

(Reviewer's comment 4) The depth of an isopycnal surface cannot be used to determine a nitracline depth. In the open ocean, nutrient concentrations are controlled by both physical processes such as vertical/horizontal advection and diffusion, convection, and biological processes such as phytoplankton growth, remineralization, etc: : : At a pinch, Eq. (2) can be used to give a crude estimation of the nitracline depth for a quasi-1D steady state system with a surface layer depleted in nutrients, with no change in solar radiation and MLD. In your case, these assumptions do not hold. Sections 3.3.3 and 3.3.3 need to be removed and the conclusion need to be changed accordingly.

(Authors' response) We fully agree that the nitracline depth is controlled by both physical and biological (i.e. depth-dependent) processes. However, actual XIXIMI bottle measurements (obtained both in summer and winter) indicate that, overall, nutrient and density vertical distribution are highly correlated (figure 6 in the submitted manuscript).

This is not something we hypothesize or suggest, nor is it new, it is what the observations indicate. Such a co-orientation of nitrate with density is not surprising, because dynamical processes that vertically displace water masses with their properties are generally strong, compared to the biological pump (Ascani et al., 2013; Omand et al., 2013 and 2015; While et al., 2010). Thus, we considered that for the time scales addressed in our analysis (i.e. seasonal), the methodology provides relevant information in the absence of direct nutrient measurements.

This methodology was applied to profiling float data by Omand et al., (2015) near the station BATS, a region characterized by the passage of mesoscale eddies and winter storms. They showed that the overall nitrate-density relationship varied relatively little throughout the year, and that the "Fluctuation in the nitracline depth due to deep mixing and mesoscale eddies (. . .) though apparent in depth coordinates (. . .) are not distinguishable in density coordinates". Note that a similar approach using an isotherm was also previously applied in the GOM to interpret the seasonal variability of surface chlorophyll (Jolliff et al., 2008). Additionally, if we look at the data at the monthly climatological scale (i.e. not profile-to-profile), the summer nitracline is generally found deeper than the winter MLD, which supports the hypothesis that large input of nutrients to the photic layer by winter mixing are likely limited at these time scales. For the above-mentioned reasons we do not consider that sections 3.3.2 and 3.3.3 need to be removed. However and as also suggested by reviewer #1 sections 3.3.2 and 3.3.3 were rewritten to make the arguments easier to follow (see our response to reviewer 1).

References

All BOEM/MMS reports can be found in: https://www.boem.gov/Gulf-of-Mexico-OCS-Region-Publications-P/#PHYSICAL OCEANOGRAPHY

Ascani, F., Richards, K. J., Firing, E., Grant, S., Johnson, K.S., Jia, Y., Lukas, R. and Karl, D.M. (2013). Physical and biological controls of nitrate concentrations in the

upper subtropical North Pacific Ocean, Deep Sea Research Part II: Topical Studies in Oceanography, 93, 119- 134.

Cox, J., Coomes, C., DiMarco, S., Donohue, K., Forristall, G.Z., Hamilton, P., Leben, R.R., and D.R. Watt, D.R..( 2010). Study of deepwater currents in the Eastern Gulf of Mexico. U.S. Dept. of the Interior, Bureau of Ocean Energy Management, Regulation, and Enforcement, Gulf of Mexico OCS Region, New Orleans, LA. OCS Study BOEMRE 2010-041. 473 pp. Donohue, K., Hamilton, P., Leaman, K. Leben, R., Prater, M., Waddell, E. and Watts. R. (2007). Exploratory Study of Deepwater Currents in the Gulf of Mexico; Volume I: Executive Summary; Volume II: Technical Report. OCS Study MMS 2006-073, 2006-074.

Donohue, K., Hamilton, P., Leben, R., Watts R., and Waddell, E., (2008). Study of Deepwater Currents in the Northwestern Gulf of Mexico Volume I: Executive Summary and Volume II: Technical Report. OCS Study MMS 2008-030 and 2008-031 Volume I - 73 pgs and Volume II - 375 p.

Donohue, K. A., Watts, D. R., Hamilton, P., Leben, R., Kennelly, M., and Lugo Fernandez, A. (2016a). Gulf of Mexico Loop Current path variability. Dynamics of Atmospheres and Oceans, 76 ( 2), 174-194. http://doi.org/10.1016/j.dynatmoce.2015.12.003.

Donohue, K. A., Watts, D. R., Hamilton, P., Leben, R., & Kennelly, M. (2016b). Loop Current Eddy formation and baroclinic instability. Dynamics of Atmospheres and Oceans, 76(2), 195-216. http://doi.org/10.1016/j.dynatmoce.2016.01.004.

Hamilton, P. and Lee, T. N. (2005). Eddies and Jets Over the Slope of the Northeast Gulf of Mexico, in Circulation in the Gulf of Mexico: Observations and Models (eds W. Sturges and A. Lugo-Fernandez), American Geophysical Union, Washington, D. C.. doi:10.1029/161GM010.

Hamilton, P., (2007). Deep-Current Variability near the Sigsbee Escarpment in the Gulf

of Mexico. Journal of Physical Oceanography, 37(3).

Hamilton, P., Lugo-Fernández, A. and Sheinbaum, J., (2016). A Loop Current experiment: Field and remote measurements. Dynamics of Atmospheres and Oceans, 76 , pp.156-173.

Hamilton, P., Bower, A., Furey, H., Leben, R.R., and Pérez-Brunius P. (2017). Deep Circulation in the Gulf of Mexico: A Lagrangian Study, U.S. Dept. of the Interior, Bureau of Ocean Energy Management, Gulf of Mexico OCS Region, New Orleans, LA. OCS Study BOEM 2017, 2017.

Jouanno, J., Ochoa, J., Pallàs-Sanz, E., Sheinbaum, J., Andrade-Canto, F., Candela, J., and Molines, J.-M. (2016). Loop Current Frontal Eddies: Formation along the Campeche Bank and Impact of Coastally Trapped Waves. Journal of Physical Oceanography, 46(11), 3339–3363. http://doi.org/10.1175/JPO-D-16-0052.1

Kolodziejczyk, N., Ochoa,J., Candela, J., and Sheinbaum, J. (2012). Observations of intermittent deep currents and eddies in the Gulf of Mexico. Journal of Geophysical Research, 117, C09014. doi:10.1029/2012JC007890, 2012.

Lewis, J. K., Kirwan, A. D., and Forristall, G. Z. (1989). Evolution of a Warm-core ring in the Gulf of Mexico: Lagrangian observations. Journal of Geophysical Research: Oceans, 94(C6), 8163-8178. http://doi.org/10.1029/JC094iC06p08163.

Omand, M. M., and Mahadevan, A. (2013). Largescale alignment of oceanic nitrate and density, Journal of Geophysical Research: Oceans, 118(10), 5322-5332.

Omand, M. M., and Mahadevan, A. (2015). The shape of the oceanic nitracline, Biogeosciences, 12(11), 3273-3287, doi:10.5194/bg-12-3273-2015.

Penduff, T., B. Barnier, W.K. Dewar, and J.J. O'Brien, (2004). Dynamical Response of the Oceanic Eddy Field to the North Atlantic Oscillation: A Model–Data Comparison. Journal of physical Oceanography, 34, 2615–2629, https://doi.org/10.1175/JPO2618.1.

Rio, M.-H., and F. Hernandez (2004). A mean dynamic topography computed over the world ocean from altimetry, in situ measurements, and a geoid model. Journal of Geophysical Research, 109, C12032, doi:10.1029/2003JC002226.

Sheinbaum, J., Badan, A., Ochoa. J., Candela, J., Rivas, D. and Gonzalez, J. I., (2007). Full-Water Column Current Observations in the Central Gulf of Mexico, Final Report. OCS Study MMS 2007-022.

While, J., and Haines, K. (2010). A comparison of the variability of biological nutrients against depth and potential density, Biogeosciences, 7(4), 1263-1269.
* * *
[Figure]

**Fig. 1.** Figure 1. Mean Dynamics topography in the Gulf of Mexico (RIO-AVISO)

[Figure]

**Fig. 2.** Figure 2. Same as fig. 5 in the submitted manuscript but for anomalies: (top) the mixed layer depth, (middle) surface chlorophyll and (bottom) integrated chlorophyll over the first 350m.

**Fig. 3.** Figure 3. Same as fig. 7 in the submitted manuscript but for anomalies: (top) the mixed layer depth, (middle) surface chlorophyll and (bottom) integrated chlorophyll over the first 350m.

---

## Author Response (AR3)

**MINOR REVISION**

Thank you for your letter of response to the reviewers and for submitting a revised manuscript. The changes you have introduced solve some of the interpretation problems highlighted by the reviewers. While the information available is not sufficient to determine unequivocally the mechanism responsible for the lack of seasonality in integrated chla concentration, your comprehensive dataset is valuable and contributes to advance our understanding of the biological oceanography of subtropical pelagic ecosystems. I am therefore recommending publication of your manuscript in Biogeosciences, subject to minor revisions. Please take into account the editorial comments below when preparing the final version of the article.

**(Authors' response)** We thank the editor for his comments and for recommending the publication of the manuscript in Biogeosciences. All editorial comments have been taken into account and are presented below in a point-by-point manner. We would also like to draw your attention to the fact that Dr. Victor F. Camacho-Ibar is now in the list of authors of the paper. All former authors consider his contribution to be important and merits his inclusion as a co-author of the paper. We hope there is no problem in doing that. The acknowledgments section was changed accordingly.

**(Editor)** p. 2 line 3 Phrase 'the GOM, as a whole, is a contrasted trophic environment' is vague. What is a 'trophic environment'? Please re-write; if what is meant is that, overall, the region is oligotrophic, just say so.

**(Authors' response)** The sentence p. 2 line 3 was rewritten "*From a biogeochemical point of view, the deep waters of the GOM are considered oligotrophic (...)*" instead of "*From a biogeochemical point of view, the GOM, as a whole, is a contrasted trophic environment. The deep basin and the continental shelf are considered oligotrophic (...)*".

**(Editor)** p.2 line 19: There are more recent studies highlighting the importance of changes in phytoplankton chlorophyll content: Behrenfeld et al. 2016 Nature Climate Change 6 323, Jakobsen and Markager (2016) Limnol. Oceanogr., 61: 1853–1868. Also the review by Halsey and Jones 2015 (Ann Rev Mar Sci) is relevant here. Note also that phytoplankton chla content changes not only in response to light, but is also sensitive to nutrient availability and temperature.

**(Authors' response)** We thank the reviewer for his suggestions. The references mentioned above were added in the manuscript, p.2 line 19.

**(Editor)** p.3 lines 4-5 What is measured here is fluorescence. Correct phrase is 'chlorophyll fluorescence', not 'fluorescence chlorophyll'

**(Authors' response)** "*fluorescence chlorophyll*" was replaced by "*chlorophyll fluorescence*".

**(Editor)** page 3, lines 12. Here authors should state which are those mechanisms.

**(Authors' response)** The sentence p.3 line 12 has been rewritten following editor's suggestions. "*(...) would be associated with a vertical redistribution of subsurface chlorophyll and/or photoacclimation processes.*" instead of "*(...) would be associated with other mechanisms described and analyzed in the following sections.*".

**(Editor)** page 3 End of Introduction: remove 'This is the most important result of our study' (not informative).

**(Authors' response)** The sentence was removed.

**(Editor)** page 4, lines 15-16. Confusing description of method. What is 'above 0.9 times the mixed layer'? The MLD is a depth, say 50 m. Then 0.9 x 50 = 45 m. Would this mean that the value of CHL at 45 m is extrapolated to the surface? Please clarify, bearing in mind that 'mixed layer' is not the same as 'mixed layer depth'.

**(Authors' response)** We agree with the editor that the sentence is unclear. What we wrote in the manuscript meant to be "mixed layer depth", not "mixed layer" and we thank the editor for noticing this confusion. Thus, p. 4, line 15-16, the sentence was rewritten: *"The method consists in finding, within the layer between the surface and 0.9 times the mixed layer depth (MLD), the highest FLUO value (FLUO$_{max}$) and its depth (FLUO$_{z,max}$). FLUO$_{max}$ is then extrapolated from FLUO$_{z,max}$ (considered as a proxy of the thickness of the layer potentially affected by the NPQ) up to the surface."* Instead of *"The method consists of extrapolating the highest FLUO value, encountered above 0.9 15 times the mixed layer, up to the surface."*.

**(Editor)** page 4 line 29 'timeS'

**(Authors' response)** Done.
* * *
**MAJOR REVISION**

Dear Editor,

Please find below our responses to the questions and comments of the reviewers. To respond, some part of the paper were rewritten and additional methodological information, discussion and references, as well as a new figure, were added, hoping they will answer both reviewers concerns and provide a better interpretation of our results. Although the results we obtained are not fully conclusive, we are convinced that they provide a significant step forward with respect to previous knowledge of the chlorophyll variability in the Gulf of Mexico (GOM). We do emphasize that our conclusions apply to basin-scale climatological averages, and they are meant to explain what is seen in a typical year in the GOM as a whole. In the following, responses to the reviewers and related changes in the manuscript are presented (the numbering of lines or pages refers to the new version of the manuscript). We do note that most of the material presented here was already sent to the open discussion.
* * *
**Reviewer #1**

**(Reviewer)** This manuscript examines estimates of the temporal and spatial distribution of chlorophyll concentration estimates in the Gulf of Mexico derived from eight profiling floats. These floats provided chlorophyll estimates by in vivo fluorescence. The authors have therefore collected a novel dataset. The primary objective result of the paper is that the spatial and temporal patterns of surface chlorophyll concentrations derived from the profiling buoys confirms the temporal and spatial chlorophyll estimates observed by ocean color satellites since

the late 1970's, and published on extensively. They confirm the seasonal deepening of the surface mixed layer depth in northern winter, in the interior of the Gulf of Mexico, and shoaling in summer. This has also been published on extensively. The causes for the seasonal surface increase and decrease in chlorophyll, and the spatial patterns defined by circulation have been also explained extensively in the literature over the past 20+ years. So, in general, the paper finds similar temporal and spatial patterns (i.e., associated with circulation features) in the Gulf of Mexico as have many other people in the past. The paper is well written in the sense that it flows well, has good prose, and is written in good English. Yet there are several problems with the paper, which may be serious enough to warrant a very deep revision and withholding publication.

(Authors' response) We thank Referee #1 for his/her review and comments. Our intention in this manuscript was to address the basin-wide seasonal average of chlorophyll concentration in the water column (not only at the surface), which is, to our knowledge, not well documented in the GOM. We agree that this data set confirms some of the main aspects of what is known about the seasonal variability of surface chlorophyll. However, this study yield the first (and to our knowledge the only one) description of the seasonal dynamics of the vertical distribution of the chlorophyll concentration ([CHL]) in the GOM, at a basin scale.

The reviewer suggests there are some serious problems in our paper. We address all the reviewer comments and criticisms in detail below, hoping they will answer the reviewer concerns and provide a better interpretation of our results.

For example, I don't understand what the authors did to compute near-surface chlorophyll concentration from the float data. They say that they took the fluorescence profile, found the highest FLUO value found above 0.9 times the mixed layer depth (MLD) and extrapolated this to the surface (as per Xing et al., 2012). They calibrate this against an ocean color satellite-derived estimate of chlorophyll concentration multiplied times 1.5 the estimated euphotic depth.

**(Authors' response)** Indeed, the first step of the procedure is to correct profiles for non-photochemical quenching (NPQ). We applied the method of Xing et al. (2012) which is actually implemented in the international BGC-Argo program and consists of extrapolating the highest fluorescence value encountered within the mixed layer up to the surface. Once the fluorescence profile is corrected from the NPQ, we determine instrumental gain and offset using ocean color satellite-derived estimates of chlorophyll concentration. For the comparison against satellite-derived estimate of chlorophyll concentration, the whole 1.5 euphotic layer is used instead of only surface records to minimize the error that would be induced by a wrong NPQ parameterization. Note that the whole procedure is described in detail in Lavigne et al., (2012) paper. So, our near-surface chlorophyll concentration estimates are based on currently accepted international standard procedures (see answer below too).

To make these points clearer, the methodological section was rewritten in page 4, from L.14 to L.19.

In addition, page 4, L.31-32, the sentence *"For the comparison, the whole 1.5 euphotic layer was used instead of only surface records to minimize the error that would be induced by a wrong NPQ parameterization."* was added.

 **(Reviewer)** One problem with this approach is that they make the same assumption of Xing et al (who did a study in the Southern Ocean) that the vertical profile of chlorophyll observed is

largely due to quenching of fluorescence, and that the deep chlorophyll maximum (DCM) is therefore not 'real'. The authors probably know that there are data collected and published since the 1960's-1970's to show that the DCM in the Gulf of Mexico is real and seasonal. I wonder if the XIXIMI-2 (July 2011) and XIXIMI-3 (February-March 2013) cruises used by the authors to obtain more than 900 water samples from 74 profiles also had some chlorophyll data? There are DNA profiles, bacterial profiles, and actual spectrophotometric and HPLC observations that show that the DCA is real and not simply an in vivo chlorophyll fluorescence quenching artifact, as the authors observed.

**(Authors' response)** We are aware that the calibration/interpretation of fluorescence measurements is critical, and this is why we paid a great deal of attention to the calibration of the data in the manuscript (section 2.2). The Xing et al. (2012) procedure has been validated and applied in various regions (e.g. BATS, HOT, DYFAMED) where a DCM is present. The relevance of the NPQ correction in these conditions/regions was specifically assessed in Lavigne et al (2012) which shows it has a positive and significant impact on the estimates of chlorophyll. That is the reason why we chose the Xing et al (2012) method for our study.

We would like to point out that we do not say that the DCM is not real or does not exist in the Gulf of Mexico. Quite the opposite, the manuscript shows how this DCM varies in concentration and depth both for the entire time series (Fig. 3) as well as seasonally in the climatological averages shown in Table 2. In fact, the NPQ correction only concerns the mixed layer and as a consequence, the impact of the correction on the fluorescence profile is generally limited to the surface and does not impact the observed DCM (except on the occasions when the mixed layer is deep enough to reach the DCM. Note that during XIXIMI cruises no HPLC observations were made.

To clarify this point, the next sentence was added, in section 2.2, page 4, L.19-20: *"The relevance of this NPQ correction in oligotrophic areas was specifically addressed in Lavigne et al., (2012) which showed it has a positive and significant impact on the estimates of chlorophyll."*.

**(Reviewer)** It is not clear to me whether the constant CHLtot seasonal cycle that they find is an artifact of the way they computed the vertical profile with the quenching correction.

**(Authors' response)** The vertically integrated chlorophyll ($[CHL]_{tot}$) depends largely on the chlorophyll content at the DCM. Hence, the contribution of the NPQ correction (which is limited to the mixed layer) is small. This can be verified in Fig. 1 below, which shows times-series from one float (float number 02 in the submitted manuscript) with and without the NPQ correction. One can see the constant [CHL]tot seasonal cycle is not an artifact of the NPQ correction.

[Figure]

Figure 1. Float 02 time-series of the (top) mean surface chlorophyll concentration and (bottom) integrated content of chlorophyll over the 0-350 m layer.

**(Reviewer)** It seems a major flaw in this paper is the conclusion that: "the present dataset reveals a vertically integrated content of chlorophyll which remains constant throughout the year, suggesting that the surface increase results from a vertical redistribution of subsurface chlorophyll or photoacclimation processes, rather than a net increase of primary productivity."

**(Authors' response)** The reviewer is absolutely right. The sentence in the abstract *"... the present dataset reveals a vertically integrated content of chlorophyll which remains constant throughout the year, suggesting that the surface increase results from a vertical redistribution of subsurface chlorophyll or photoacclimation processes, rather than a net increase of primary productivity"* is wrong. We thank the reviewer for noticing that, and the term primary production was changed to biomass, which is what we actually meant and missed to correct before submitting. Note that this was correctly stated in our conclusion number 3.

The sentence in the abstract, page 1, L.17-19, is now: *"... the present dataset suggests that the basin scale climatological surface increase in chlorophyll content results from a vertical redistribution of subsurface chlorophyll and/or photoacclimation processes, rather than a net increase of biomass."*

**(Reviewer)** The problem is that the integrated water column productivity of a water column with a DCM is not the same as that same water column under a "spring bloom" condition, when phytoplankton biomass is high throughout the mixed layer. The literature is replete with actual measurements of primary productivity that show this. In my opinion, the ecological and biogeochemical interpretation that biomass is the same as productivity is a fatal flaws for this paper. The authors need to go back and fully investigate what mixing can do to phytoplankton blooming in the ocean. They need to review what chlorophyll represents (a crude index of biomass), what productivity is (a rate), and what other factors may play a role in changing these over time and space.

**(Authors' response)** We fully agree with the reviewer that chlorophyll is only an index of biomass which is not the same as productivity and regret some confusions made in the manuscript. In the revised version, we are careful to describe fluorescence measurements as a

proxy of chlorophyll concentrations, and then to discuss these data (with respect to backscattering measurements) solely in terms of biomass. Primary production (and more precisely new primary production), is now evoked only when nutrients fluxes are estimated (section 3.3), with the aim to discuss it as a hypothesis and/or a possible mechanism (not as a direct result of chlorophyll measurements).

In order to make these points clearer, substantial changes were made in the new version of the manuscript:

The section 3.1 was deeply rewritten from page 7, L.28 to page 8, L.28, and, in particular, the next sentences were added (from page 7, L.32-33 to page 8, L.1-2) to stress what chlorophyll represents *"... one may wonder how much of the phytoplankton chlorophyll variability is reflective of true changes in total biomass in the entire water column. Indeed, it is well known that the [CHL] is not a sole function of phytoplankton biomass and depends on several other factors, such as photoacclimation processes (e.g. Geider, 1987).".*

Section 3.3.2 page 11, L.9, *"... does not necessarily reflect a real increase ..."* Instead of *"... does not reflect a real increase ...".*

Section 3.3.2, sentences *"This is in full agreement with results obtained from [CHL]. Thus, the idea that winter production in the GOM is enhanced in winter by new nutrients availability may be a misconception."* were removed.

Section 3.3.3, page 12, L. 3 *"…a higher chlorophyll concentration was measured… "* instead of *"... a higher biomass concentration was measured…".*

Section 3.3.3 page 12, L.18 *"… the observed enhanced [CHL] and biomass …"* instead of *"… the observed enhanced biomass …".*

Given that it implies of lot of suppositions and that it does not add significant information in Section 3.3.3. the sentence *"Nevertheless, we can note that the estimated NPP in CG is higher than in AG by a factor 1.13 ±0.02, on average, which is surprisingly close to the mean [CHL]tot ratio between CG and AG (1.15 ± 0.08)."* was removed.

Conclusion number 4, the sentence *"This result stands in contradiction with the current paradigm of an enhanced primary production in winter, triggered by nutrient input through vertical mixing ...."* was removed.

Conclusion number 8 page 13, L.27, *"the mechanisms controlling biomass variability and primary production in the GOM."* instead of *"...the mechanisms controlling primary production in the GOM."*

 **(Reviewer)** What is amazing is that the authors consider past biological oceanographic studies and conclusions of observations in the Gulf of Mexico to be 'beliefs', and proceed to completely misinterpret the chlorophyll signal they observe.

**(Authors' response)** We agree the term is not appropriate. We were trying to emphasize the fact that, in the GOM, the seasonal cycle of chlorophyll, at a basin scale, has been almost exclusively addressed using satellite measurements, which only provide surface information.

Sentences in questions were rewritten to make clear we are just comparing our results and hypothesis with others studies, not disregard them.

In the abstract, page 1, L.16-17 "*a possible interpretation*" instead of "*current belief*".

In the introduction, page 2, L.13 "*it has been suggested*" instead of "*it is currently thought*".

In the introduction, page 3, L.10-12 "*Our analysis indicates that at a basin scale, the winter surface [CHL] maximum in the GOM may not necessarily be produced by a biomass increase, but would be associated with other mechanisms ...*" instead of "*In contrast to common belief, our analysis indicates the winter surface [CHL] maximum in the GOM is not produced by a biomass increase, but by other mechanisms ...*".

(**Reviewer**) They interpret their observations to mean that there are no water-column integrated changes in chlorophyll AND in primary productivity in the Gulf of Mexico. This is clearly a gross misinterpretation of the crude biomass index data they collected. The authors did not exploit the data to make inferences on primary productivity (e.g. perhaps by looking at hour-to-hour and day-to-day changes in biomass). The authors should note that estimates of primary productivity and of chlorophyll concentration are also out of phase in time in the Gulf of Mexico. This has also been reviewed in the literature.

(**Authors' response**) As we answered above, we agree that we can only address biomass with chlorophyll data, and modifications have been done to make it clearer in the manuscript (see comment above). We indeed observed that the integrated content of chlorophyll does not show a clear seasonal variability, which we interpret as total biomass remaining constant throughout the year at monthly timescales (which is consistent with the analysis of the bbp data shown in the supplementary material). In addition, the temporal resolution of our floats measurements (14 days) prevented us to infer hour-to-hour and day-to-day changes in biomass to estimate in an appropriate manner primary production.

(**Reviewer**) Another problem is the interpretation of nutrient data. The authors have a rich nutrient dataset with the density data computed from the buoy profiles and the nutrient data from the XIXIMI-2 (July 2011) and XIXIMI-3 (February-March 2013) cruises. The analysis of the density vs. nutrient data is very nice. The problem starts when the authors interpret the nutrient profiles in a biogeochemical and ecological manner. They assume that simply because we see a winter-time increase in chlorophyll concentration in the mixed layer, there also needs to be a clear, measureable signal in nutrient concentrations. Since they don't see this, they conclude that "there are no significant inputs of nutrients by vertical mixing to sustain significant winter new primary production (NPP)". This is incorrect. Nutrients will not be measurable as they are taken up by the phytoplankton. This has been published over and over in the course of the past half century or longer.

(**Authors' response**) We agree with the reviewer's comment that "Nutrients will not be measurable as they are taken up by the phytoplankton" (see e.g. page 11, L.19-20 in the manuscript). To make this point clearer, we made the following modifications:

Section 3.1 page 8, L.26-28, "*the mixed layer in winter, although sufficiently deep to reach the DCM, would be, nonetheless, insufficient for bringing up large quantities of nutrients and support a significant net increase in phytoplankton biomass*" Instead of "*the mixed layer in*

*winter is sufficiently deep to reach the DCM but nonetheless insufficient for bringing up nutrients".*

Section 3.3.2, page 11, L.11-15, was rewritten "... $Z_N$ *is always found at depth and does not show a clear seasonal pattern (regardless of the group). In addition, the climatological winter mixed layer is generally shallower than the nitracline (Fig. 5). Hence if we assume that large inputs of nutrients can only be expected when the MLD reaches below the average nitracline depth ($Z_N$), it is likely that nutrients injections to the photic layer by vertical mixing are low on average, even in winter.*" instead of "... $Z_N$ *is always found at depth (regardless of the group) and never approaches the surface, even in winter. This result means that there is no NN accumulation in surface waters and that the deep nutrient reservoir is always isolated from the surface layer ...".*

Section 3.3.2, page 11, L.21-23, *"Thus, apart from sporadic and rather localized events, it seems likely that large supplies of nutrients to the surface layer are not that common in winter in the GOM as a whole, since the basin scale, monthly climatological basin-scale averages of the MLD are shallower than the estimated depth of the nitracline"* Instead of *"it is likely that on average, there are no significant inputs of nutrients by vertical mixing to sustain significant winter new primary production (NPP)."*

Section 3.3.2, sentences *"This is in full agreement with results obtained from [CHL]. Thus, the idea that winter production in the GOM is enhanced in winter by new nutrients availability may be a misconception."* are removed.

Conclusion number 4, page 13, L5-11 was rewritten: "*In addition, our observations show that the winter mixed layer is generally not deep enough to reach the nitracline. The sampling, however, only allows to reach conclusions in a broad sense. Therefore, we suggest that, on a climatological basin-scale average, a relatively small amount of nutrients are potentially injected to the surface layer through vertical mixing. This does not discard the fact that at short time-scales (days to weeks), events may result in high nutrient inputs to the photic layer which translate in a local phytoplankton bloom, particularly during winter storms. Our interpretation is that the net effect of those blooms is not big enough to determine the basin scale averages of surface chlorophyll content, hence nutrient supply by winter mixing is not necessarily the main cause of the seasonal, basin-scale variability of surface chlorophyll content.*".

**(Reviewer)** The authors seem to somehow dismiss biological oceanography theory in general, including historical knowledge of patterns of vertical distribution of chlorophyll concentration, how these vary in time, and how all this and oceanographic conditions (both biotic and abiotic) affect primary productivity.

We have not found literature which addresses the seasonal variability of the vertical distribution of chlorophyll in the Gulf of Mexico. Indeed, in the 90s Muller- Karger et al., (1991) stated: *"(...) in situ oceanographic data set for the Gulf of Mexico is still insufficient to address questions and processes affecting the distribution of biological and chemical properties."* and recently (Muller-Karger et al., 2015): "*The waters of the interior of the Gulf of Mexico seaward of the continental margin continue to be seriously undersampled. (...). We were not able to derive a good chlorophyll concentration dataset from historical field observations archived at the NOAA NODC to compare with either CZCS, SeaWiFS, or MODIS chlorophyll estimates. Most samples in the Gulf of Mexico available at the NODC are from the northern and eastern shelf regions, with relatively few samples available from offshore waters. Thus, information derived*

*from remote sensing is essential for characterization of the deep water areas of the Gulf.*"
Moreover, we are not aware of studies showing mixed layer depth climatologies based on direct
measurements for the Gulf of Mexico either, since the ones available are indirect estimates
based on models or parametrizations (Mandoza et al., 2005; Muller-Karger et al., 2015; Zavala-
Hidalgo et al., 2014).

In this paper we specifically focused on the measured MLD and considered the ecosystem from
a bottom-up perspective. We agree that biomass is regulated by a wider range of processes (e.g.
biotic processes) which are not directly addressed in our manuscript since to answer these
questions would require other measurements that were not at our disposal. This is a limitation
to our work which we were careful to mention in the manuscript (section 3.3.3 page 12 lines
20-23, and conclusion 7 page 13, L.9-13). Hence, we consider that the new measurements of
water column chlorophyll and bpp seasonal variability invite to new hypotheses that are worth
exploring.

To clarify, a conceptual diagram has been added in the revised manuscript (Fig. 8) to outline
the processes we are addressing (MLD and mesoscale) and their impact on the vertical
distribution of the [CHL] in the GOM.

**(Reviewer)** The reference: Heileman, S., and Rabalais, 2009, cited to provide a reference on
the productivity of the Gulf of Mexico is not a reference for the characterization of productivity
in the Gulf. It does not provide summary data. The authors should cite where the actual
productivity data comes from that they use to characterize productivity in the Gulf of Mexico.

**(Authors' response)** In the new version of the manuscript, the reference Heilman and Rabalais
(2009) is not used anymore to characterize primary production in the GoM.

The paragraph in the introduction, page 2 L.3-8 was partially rewritten and two references (with
the corresponding estimates of primary production) were added: "*From a biogeochemical point
of view, the GOM, as a whole, is a contrasted trophic environment. The deep basin and the
continental shelf are considered oligotrophic and nutrient-limited being relatively isolated from
coastal and eutrophic waters (Heileman and Rabalais, 2009). In deepwater GOM, historical
in situ measurements indicate low biological productivity (< 150 mgC m$^{-2}$ d$^{-1}$) and low surface
chlorophyll concentration (hereafter [CHL]$_{surf}$) with values ranging from 0.06 to 0.32 mg m$^{-3}$,
and being 2-3 times higher in subsurface waters (Biggs and Ressler, 2001; El-Sayed, 1972;
Koblenz-Mishke et al., 1970).*" Instead of "*From a biogeochemical point of view, the GOM, as
a whole, is a moderately high productivity ecosystem (< 300 gC m$^{-2}$ yr$^{-1}$) with a contrasting
trophic environment (Heileman and Rabalais, 2009). The deep basin and the continental shelf
are considered oligotrophic and nutrient-limited being relatively isolated from coastal and
eutrophic waters (Heileman and Rabalais, 2009). In situ measurements indicate surface
chlorophyll concentration (hereafter [CHL]$_{surf}$) ranging from 0.06 to 0.32 mg m$^{-3}$, with [CHL]
2-3 times higher in subsurface waters (Biggs and Ressler, 2001).*"

**(Reviewer)** The authors do this often– they cite relatively recent references (in the decade of
the 2000's). When they cite earlier literature, they do this in passing and in a dismissive manner,
not fully acknowledging that many of the points treated in this paper has already been discussed
and explained previously. The problem is that, in doing this, they miss important background
knowledge about the oceanography of the Gulf of Mexico.

(Authors' response) A total of 9 references were added in the introduction (see response above) and in section 2.5 and 3.2.

(Reviewer) Also, the authors cite studies by Behrenfeld et al (2005), Mignot et al (2014), etc. as suggesting that all temporal changes in chlorophyll observed by satellite are due to changes in pigment concentration in phytoplankton cells. This may be part of what happens, but it is not an accurate characterization of the changes that occur in the Gulf that they measured.

(Authors' response) We agree with the reviewer that the observed changes may not only be due to photoacclimation, and we do not state that this is the only relevant processes involved (see our discussion in section 3.1, page 8, L9-25, and our conclusion number 7). However, this new dataset suggests that photoacclimation may be relevant and worth exploring. The above references of Behrenfeld et al., (2005), Mignot et al., (2014) are in the discussion to support this hypothesis.

To make it clearer, following changes were made:

Section 3.1 page 8, the section was deeply rewritten (see our response above). In particular the discussion about the relevance of the different mechanisms that could lead to a winter $[CHL]_{surf}$ increase without a similar trend in $[CHL]_{tot}$, was developed (page 8, from L.9 to L.28).
* * *
**Reviewer #2**

(Reviewer) The manuscript Temporal variability of chlorophyll distribution in the Gulf of Mexico: bio-optical data from profiling floats by Pasqueron De Fommervault et al. attempts to evaluate the temporal and spatial variability of chlorophyll concentration in the Gulf of Mexico. The study utilises data from eight bio-optical profiling floats. The paper addresses the winter increase in sea surface chlorophyll concentration and the impact of mesoscale eddies on phytoplankton biomass in the Gulf of Mexico. This is done by applying already published methods to a rather new dataset. The problem is that, most of the time, the methods cannot be applied or are not applied correctly. Consequently, most of the results presented in the manuscript are flawed. Additionally, the manuscript is poorly written, the arguments are hard to follow, and often not justified by the appropriate references. The amount of work needed for publication in Biogeosciences is considerably more than for a major revision.

(Authors' response) We thank Referee #2 for his/her review and comments. The major criticisms are related to methodological issues that prevent, according to the reviewer, an accurate analysis of our dataset. Below, we address and discuss all the comments hoping they will clarify our work and answer the reviewers' concerns. Additional references and figures are also presented. The comment that the manuscript is poorly written and the arguments hard to follow is difficult to address since no specifics are provided. We however added a conceptual diagram in the revised manuscript (Fig. 8) hoping it will make the arguments easier to follow.

One of the conclusion of the manuscript is that the winter increase in sea surface chlorophyll concentration is due a photoacclimation process or a reentrainment of phytoplankton cells at depths. I felt like the authors have chosen to copout on testing one hypothesis over another. The authors have all the data necessary to investigate what causes the winter increase in sea surface chlorophyll concentration Gulf of Mexico. They need to do more than just commenting on the float observations.

**(Authors' response)** In the manuscript it is suggested that the winter surface chlorophyll increase could result from (i) photoacclimation within the mixed layer (ML) due to a reduction in average light availability, and/or (ii) from mixing of the deep chlorophyll maximum (DCM) into the ML. Processes involved in (i) are considered on a seasonal basis. The hypothesis is that during the winter mixing period, the average light intensity for phytoplankton is reduced in the ML with respect to summer conditions leading to an increase of intracellular chlorophyll content. Since, on average, a low bbp/[chl] ratio at the surface is persistent in winter, this hypothesis seems reasonable to explain the variability observed in surface. Hypothesis (ii) supposes that low-light acclimated cells from the DCM are transported to the surface by mixing process, thus increasing the bbp/[chl] ratio at the surface. Considering only (ii) assumes that phytoplankton cells do not have time to re-acclimate to their new light environment (hours to days processes). So it is not warranted to test (ii), given the temporal resolution of our floats (two profiles per month from each float. We do not think the data have enough temporal resolution to determine what causes the winter increase in surface chlorophyll. That is the reason why we are cautious and suggest that is difficult to attribute the observed changes in chlorophyll to only one mechanism and suggest/speculate possible mechanisms. However we fully agree with the reviewer that this point would require a deeper discussion.

Accordingly, all the section 3.1 was rewritten and, in particular, the discussion about the relevance of both mechanisms (page 8, L.9-28) was detailed.

In addition in the abstract, page 1, L. 19 "… *a vertical redistribution of subsurface chlorophyll and/or photoacclimation processes.*" Instead of "*... a vertical redistribution of subsurface chlorophyll or photoacclimation processes.*".

In the conclusion number 3, page 13, L. 4 "*...a vertical redistribution of chlorophyll and/or photoacclimation processes, rather than a true biomass increase.*" Instead of "*...a vertical redistribution of chlorophyll or photoacclimation processes.*".

**(Reviewer)** The authors use the depth of the 6C isotherm to classify the eddies in the Gulf of Mexico. They argued that this isotherm has a mean depth of 795 m and that Bunge et al (2002) found that this isotherm separates the deep stable water from the eddy-influenced surface water. Finally, it is said that Hamilton et al. 2017 found a strong correlation between the isotherm and upper layer eddies. First, Bunge et al. (2002) did not identify eddies with this isotherm. It was used to delineate the depth of the Loop Current in the Yucatan channel. Consequently, the use of the 6C isotherm to identify mesoscale eddies cannot be justify by this reference. Second, the vertical extent of eddies core is comprised, in average, between 300 and 400 m. What is the rationale of using such a deep isotherm to detect eddies that impact the first 400m of the water column? Third, the authors claimed that Hamilton et al. (2017) found a correlation between mesoscale eddies and the 6C isotherm. However, this reference seems to be an oral presentation and again it cannot be used to justify the utilization of the 6C isotherm.

**(Authors' response)** First, note that the analysis is meant to detect mesoscale structures (*i.e* nearly geostrophic flow structures that can be identified from sea surface height anomalies / thermocline depth anomalies) but also structures that are part of the mean circulation. In the Gulf of Mexico (GOM), besides transient eddies such as the anticyclones detached from the Loop Current (LC), the cyclones that often accompany the detachment process and other eddies present inside the GOM, there are semi-permanent structures that, for example, can be seen in the figure below (Figure 2) which shows the mean dynamic topography of the GOM (Rio and

Hernandez, 2004, and mean-dynamic topography https://www.aviso.altimetry.fr/en/applications/ocean/large-scale-circulation/mean-dynamic-topography.html). One can see that the LC itself (anticyclonic), the cyclonic gyre of the Bay of Campeche and the (anticyclonic) eddy/gyre straddling the central GOM (produced in part by the path of LC eddies inside the GOM) are features of the mean circulation.

[Figure]

Figure 2. Mean Dynamics topography in the Gulf of Mexico (RIO-AVISO).

Hence, to avoid misunderstanding and to make sure it is clear we are referring to mesoscale structures and not only "eddies" which are generally understood as transient and closed circulation features, we have made the following changes:

Section 2.5, from page 5, L.31 to page 6, L. 1 "*The classification of the mesoscale structures (which, in the GOM, encompass eddies but also structures that are part of the mean circulation such as LC and LC eddies), was carried out using the depth of the 6°C-isotherm…*"instead of "*The classification of the eddy structures was carried out using the depth of the 6°C isotherm…*".

Section 2.5, page 6, L.23 the term *eddy* is now replaced by *mesoscale*.

Section 2.5 the sentence "*One should note that our determination of mesoscale anticyclonic structures encompasses both the eddies and the Loop Current*" was removed.

Section 3.3.1, page 10, L.12 "*the two groups*" instead of "*the two eddy groups*".

Second, in several references (e.g. Donohue et al., 2007; Hamilton 2007; Sheinbaum et al., 2007; Donohue et al., 2008; Hamilton et al., 2016) it is shown that the vertical displacement of the 6 to 10∘C isotherms (located at the base of the LC and LC eddies) is anti-correlated with sea-surface height displacements, highlighting the importance of first baroclinic mode dynamics in the GOM. 6-7∘C isotherms are at 450-500 m depth within some cyclonic eddies. Donohue et al., (2008), Cox et al., (2010) and Hamilton et al., (2016) suggest the 6∘C isotherm is a better choice for a two-layer analysis, and Kolodziejczyk et al., (2012) find the zero crossing of the first baroclinic mode near 1000 m depth.

We agree with the reviewer that not enough references or explanations for using the 6°C isotherm depth (T6) variability used to identify mesoscale structures were given in the submitted manuscript. Thus, to make a stronger case for the use of T6 to identify mesoscale structures, 5 references were added (Section 2.5, page6, L.6, 7, 10 and 11) and list items of pages 6 (L.4-12) were rewritten:

- *The GOM can be studied as a two-layer baroclinic system, with vertical displacements of the 6 to 10°C isotherms (located at the base of the LC and LC eddies and at 400-600 m depth in cyclonic eddies) being anti-correlated with sea-surface height (Donohue et al., 2007 and 2008; Hamilton 2007b, 2016a and 2016b; Sheinbaum et al., 2007).*

- *T6 is in the lower thermocline and has been used as the interface for a two layer system, separating deep waters from the more energetic upper layer containing mesoscale structures and the LC (Bunge et al., 2002; Donohue et al., 2008; Hamilton et al. 2016a and 2016b).*

- *T6 is directly measured by the profiling floats.*

Third, the depth of T6 is directly measured by the floats. Hence we decided to use this parameter to track mesoscale structures instead of sea surface height (SSH) measured by satellites, which have limited spatial resolution so the required interpolation to the locations of our profiles would add additional errors. This is discussed in the manuscript (page 6, Line 26-33) and the figure S1 (supplementary material of the manuscript), show that using SSH instead of T6 leads to essentially the same results.

Finally, note that the baroclinic signal of the mesoscale structures reaches down below 600m (or more) in the GOM (e.g. Lewis et al., 1989; Donohue 2016a and b, Sheinbaum et al., 2007, Hamilton et al., 2016a), even if the eddies' core is shallower. Additionally, Hamilton et al., (2016a) reference given in the manuscript is not an oral presentation (it is a BOEM-MMS technical report which is available online (https://www.boem.gov/ESPIS/5/5583.pdf). The link is now included in the reference list.

Given the above, we decided to leave the analysis as is, using the T6 criteria to obtain the anticyclonic/cyclonic subsets.

**(Reviewer)** The section on the impact of mesoscale eddies on phytoplankton biomass is largely inspired by the study of Dufois et al. 2014. The authors compared the float observations when the floats were profiling in a cyclonic structure with the observations when the floats were profiling in an anticyclonic structure. By averaging observations that were collected at different times of the year and locations in the Gulf of Mexico, you are looking at signals that are both influenced by the seasonal and large scale variability and the mesoscale activity; this is not correct. To properly assess the impact of eddy signal on a given variable, one needs to look at the departure from the seasonal mean (see Cushman-Rosin, 1994). All sections about the role of mesoscale structures need to be changed.

(Authors' response) We did not "average observations that were collected at different times of the year and locations in the Gulf of Mexico" as the referee writes, since we did not average different times of the year. We clustered the data in different groups based on identifying whether the profiles were taken in anticyclonic, cyclonic or neutral vorticity anomalies,

following the criteria explained in the text (see also comment above), and then simply computed monthly climatologies for the anticyclonic and cyclonic subsets, gathering all profiles corresponding to each month regardless of the year and the geographic position, and looked for differences between these two clusters. Data are evenly distributed so each month of the year has a similar number of profiles (figure 1, right panel in the manuscript). One may question the representativeness of this climatology given the amount of data (two vertical profiles per month during 5 years from seven APEX floats distributed unevenly in the GOM), but it is what is available now and sufficient, in our view, to make this computation worthwhile.

It is well known that the seasonal cycle of geophysical and biogeochemical variables is not monochromatic and has time scales that overlap those of mesoscale turbulence. Both processes are thus impossible to separate properly, "even by defining running variances over 6-month windows or by subtracting a mean seasonal cycle from the time series before variance computations" (Penduff et al., 2004). So a decomposition of a variable as a time-mean+seasonal+"eddy" is not necessarily warranted as a means to separate seasonal and mesoscale eddy signals. Besides, depending on how calculations are made, the "eddy" part will include lower frequency variations too. Although there is certainly seasonal variability in the GOM related to atmospheric forcing (air-sea interaction river run-off), spectra (e.g. SSH) shows comparable or more energetic variations related to the mesoscale, in a band of frequencies that overlaps with seasonal variations (3-4 months, Jouanno et al, 2016; Hamilton and Lee, 2005).

Having said that, we followed the suggestion of the reviewer to first remove the seasonal signal to try to separate it from the mesoscale variability. We first calculate the climatology for the variable in question using all profiles, and then substract the monthly mean value from each profile to compute the anomalies. These anomalies have now mesoscale as well as lower frequency (interannual) signals. The profiles are then separated in the two clusters (cyclonic and anticyclonic), and a monhtly climatology of the anomalies is obtained. Figures below (Figures 3 and 4), same as figure 5 and figure 7 in the manuscript but for anomalies, show the same results as those obtained from the original calculation. Namely that anticyclones tend to show a deeper mixed layer depth (MLD) than cyclones in winter, that cyclones have slighlty larger integrated chlorophyll content than anticyclones, and that the nitracline depth and steepness is larger for cyclones throughout the year.

We decided not to use the anomalies, given that they are only useful to show relative differences between the two groups (anticyclonic vs cyclonic) of a given parameter, but it becomes complicated when two parameters are to be compared (e.g. the depth of the ML with that of the DCM). We think the discussion is clearer if we stick to the absolute values, and given the results above, we hope that the reviewer agrees with us.

[Figure]

Figure 3. Same as fig. 5 in the submitted manuscript but for anomalies: (top) the mixed layer depth, (middle) surface chlorophyll and (bottom) integrated chlorophyll over the first 350m.

[Figure]

Figure 4. Same as fig. 7 in the submitted manuscript but for anomalies: (top) the mixed layer depth, (middle) surface chlorophyll and (bottom) integrated chlorophyll over the first 350m.

Given the above, we made the following changes in the section 3.2:

Page 9, L.1 the title was changed: "*Impact of mesoscale structures on the annual cycle*" instead of "*Role of mesoscale structures in shaping the annual cycle*".

Page 9, L.12-19, The next paragraph was added: "*It is well known that the seasonal cycle of geophysical and biogeochemical variables is not monochromatic and has time scales that overlap those of mesoscale turbulence (e.g. Penduff et al., 2004) thus, it is not possible to separate them properly. Although there is certainly seasonal variability in the GOM related to atmospheric forcing (air-sea fluxes and river run-off), spectra shows comparable or more energetic variations related to the mesoscale in a band of frequencies that overlaps with seasonal variations (3-4 months for sea surface height, e.g., Hamilton and Lee, 2005; Jouanno et al, 2016). Hence, it could be expected that the variability observed at a given period of the year also depends on the presence of mesoscale structures. We therefore analysed the seasonal cycle gathering data on a monthly-basis and considering separately profiles acquired in cyclonic and anticylonic structures, to assess MLD and [CHL] ([CHL]$_{surf}$ and [CHL]$_{tot}$) differences between the two groups.*".

Page 9, L.20, "*climatological basin-scale averaged MLD*" instead of "*MLD*".

Page 9, L.29, "*The monthly climatological mean [CHL]$_{tot}$*" instead of "*Monthly mean [CHL]tot*".

Legend of figure 5 and 7 were also modified:

Figure 5: "*Basin-scale, monthly climatological mean and standard deviation of: (top) the mixed layer depth, (middle) surface chlorophyll and (bottom) integrated chlorophyll over the first 350m. Red (blue) shows statistics for all profiles in anticyclonic (cyclonic) structures.*" Instead of "*Monthly mean and standard deviation of: (top) the mixed layer depth, (middle) surface chlorophyll and (bottom) integrated chlorophyll over the first 350m.*".

Figure 5: "*Basin-scale, monthly climatological mean and standard deviation of: (top) the nitracline depth, (bottom) the nitracline steepness. Red (blue) shows statistics for all profiles in anticyclonic (cyclonic) structures. Red (blue) shows statistics for all profiles in anticyclonic (cyclonic) structures.*" Instead of "*Monthly mean and standard deviation of: (top) (top) the nitracline depth, (bottom) the nitracline steepness.*".

**(Reviewer)** The depth of an isopycnal surface cannot be used to determine a nitracline depth. In the open ocean, nutrient concentrations are controlled by both physical processes such as vertical/horizontal advection and diffusion, convection, and biological processes such as phytoplankton growth, remineralization, etc: : : At a pinch, Eq. (2) can be used to give a crude estimation of the nitracline depth for a quasi-1D steady state system with a surface layer depleted in nutrients, with no change in solar radiation and MLD. In your case, these assumptions do not hold. Sections 3.3.3 and 3.3.3 need to be removed and the conclusion need to be changed accordingly.

**(Authors' response)** We fully agree that the nitracline depth is controlled by both physical and biological (i.e. depth-dependent) processes. However, actual XIXIMI bottle measurements (obtained both in summer and winter) indicate that, overall, nutrient and density vertical distribution are highly correlated (figure 6 in the submitted manuscript). This is not something

we hypothesize or suggest, nor is it new, it is what the observations indicate. Such a co-orientation of nitrate with density is not surprising, because dynamical processes that vertically displace water masses with their properties are generally strong, compared to the biological pump (Ascani et al., 2013; Omand et al., 2013 and 2015; While et al., 2010). Thus, we considered that, for the time scales addressed in our analysis (*i.e.* seasonal), the methodology provides relevant information in the absence of direct nutrient measurements.

This methodology was applied to profiling float data by Omand et al., (2015) near the station BATS, a region characterized by the passage of mesoscale eddies and winter storms. They showed that the overall nitrate-density relationship varied relatively little throughout the year, and that the "*Fluctuation in the nitracline depth due to deep mixing and mesoscale eddies (. . .) though apparent in depth coordinates (. . .) are not distinguishable in density coordinates*". Note that a similar approach using an isotherm was also previously applied in the GOM to interpret the seasonal variability of surface chlorophyll (Jolliff et al., 2008). Additionally, if we look at the data at the monthly climatological scale (i.e. not profile-to-profile), the summer nitracline is generally found deeper than the winter MLD, which supports the hypothesis that large input of nutrients to the photic layer by winter mixing are likely limited at these time scales. For the above-mentioned reasons we do not consider that sections 3.3.2 and 3.3.3 need to be removed. However and as also suggested by reviewer #1 changes in sections 3.3 were realized to make the arguments easier to follow (see our response to reviewer 1), and conclusion 4 rewritten.

In particular:

Section 3.3.2, page 11, L.11-15, was rewritten "... $Z_N$ is always found at depth and does not show a clear seasonal pattern (regardless of the group). In addition, the climatological winter mixed layer is generally shallower than the nitracline (Fig. 5). Hence if we assume that large inputs of nutrients can only be expected when the MLD reaches below the average nitracline depth ($Z_N$), it is likely that nutrients injections to the photic layer by vertical mixing are low on average, even in winter." instead of "... $Z_N$ is always found at depth (regardless of the group) and never approaches the surface, even in winter. This result means that there is no NN accumulation in surface waters and that the deep nutrient reservoir is always isolated from the surface layer ...".

Section 3.3.2, page 11, L.21-23, the sentence "*Thus, apart from sporadic and rather localized events, it seems likely that large supplies of nutrients to the surface layer are not that common in winter in the GOM as a whole, since the basin scale, monthly climatological basin-scale averages of the MLD are shallower than the estimated depth of the nitracline.*" was added.

*Conclusion number 4, page 13, L5-11 was rewritten: "In addition, our observations show that the winter mixed layer is generally not deep enough to reach the nitracline. The sampling, however, only allows to reach conclusions in a broad sense. Therefore, we suggest that, on a climatological basin-scale average, a relatively small amount of nutrients are potentially injected to the surface layer through vertical mixing. This does not discard the fact that at short time-scales (days to weeks), events may result in high nutrient inputs to the photic layer which translate in a local phytoplankton bloom, particularly during winter storms. Our interpretation is that the net effect of those blooms is not big enough to determine the basin scale averages of surface chlorophyll content, hence nutrient supply by winter mixing is not necessarily the main cause of the seasonal, basin-scale variability of surface chlorophyll content.*".

We would like to mention that the biogeochemical model study by Damien *et al.,* (submitted to Journal of Geophysical Research- Ocean) confirms that a winter MLD deeper than the nitracline (and a subsequent input of nutrients to the surface layer), is not a common feature in the Gulf of Mexico. This reference was added in the revised manuscript and the next sentence added in the conclusion number 9 (page 13, L. 23-27): *"The model results are consistent with the hypothesis stated in this work, but also highlights that the BOEM floats' sampling scheme is unable to resolve all the scales of temporal and spatial variability."*
* * *
*LIST OF RELEVANT CHANGES*

- Dr. Victor F. Camacho-Ibar was added in the list of authors of the paper.
- 14 references were added.
- 1 figure was added (Figure 8).
- Methodological section was rewritten p. 4 L. 12-20 "Calibration of fluorescence profiles" and p.6 L. 4-12 "Detection of mesoscale structures".
- Section 3.1 "Seasonal cycle" was entirely rewritten and a discussion was added.
- Section 3.2 "Impact of mesoscale structures on the annual cycle". A discussion was added p.9 L. 9-16.
- Section 3.3.2 "3.3.2 Winter mixing" was rewritten p.11 L.7-12 and p.11 L.16-19.
- Conclusion number 4 was rewritten.
* * *
*MARKED-UP MANUSCRIPT VERSION*
* * *

[revised manuscript text omitted]